# Mapping functional diversity from remotely sensed morphological and physiological forest traits

Fabian D. Schneider [1], Felix Morsdorf [1], Bernhard Schmid [2], Owen L. Petchey [2], Andreas Hueni [1], David S. Schimel [3] & Michael E. Schaepman [1]

Assessing functional diversity from space can help predict productivity and stability of forest ecosystems at global scale using biodiversity–ecosystem functioning relationships. We present a new spatially continuous method to map regional patterns of tree functional diversity using combined laser scanning and imaging spectroscopy. The method does not require prior taxonomic information and integrates variation in plant functional traits between and within plant species. We compare our method with leaf-level field measurements and species-level plot inventory data and find reasonable agreement. Morphological and physiological diversity show consistent change with topography and soil, with low functional richness at a mountain ridge under specific environmental conditions. Overall, functional richness follows a logarithmic increase with area, whereas divergence and evenness are scale invariant. By mapping diversity at scales of individual trees to whole communities we demonstrate the potential of assessing functional diversity from space, providing a pathway only limited by technological advances and not by methodology.

---

[1] Remote Sensing Laboratories, Department of Geography, University of Zurich, Winterthurerstrasse 190, CH-8057 Zurich, Switzerland. [2] Department of Evolutionary Biology and Environmental Studies, University of Zurich, Winterthurerstrasse 190, CH-8057 Zurich, Switzerland. [3] Jet Propulsion Laboratory, California Institute of Technology, 4800 Oak Grove Drive, Pasadena, CA 91011, USA. Correspondence and requests for materials should be addressed to F.D.S. (email: fabian-daniel.schneider@geo.uzh.ch)

Understanding community structure and the impact of changing biodiversity on ecosystem functioning are key tasks in ecology. Progress has been made on a wide variety of taxa, including plants[1], fish[2], birds[3] and insects[4], amongst others. In plant ecology, biodiversity research has focused on the distribution of species based on taxonomic identity[5]. More recently, with the emergence of functional biogeography[6], tree species or individuals of a community are described in relation to their functional identity and distribution in space. Functional traits are of particular interest due to their response to environmental conditions and direct link to growth, reproduction and survival[7, 8]. Trait-based approaches are emerging rapidly in plant ecology, underpinning community assembly and structure, species interactions and interlinkages between vegetation and biogeochemical cycles[9].

The assessment of plant functional traits and plant functional diversity is of particular relevance when predicting ecosystem productivity and stability. A multitude of experimental studies demonstrated positive relationships between plant diversity and ecosystem functioning[10–12] and increasingly such positive relationships are also found in comparative observational studies[13, 14]. A positive relationship over extended time scales is mainly driven by functional diversity due to an increased resource use efficiency and utilization as well as sampling effects in a changing environment, allowing plant communities to sustain high productivity over time[15–17]. Besides productivity, higher functional diversity has been linked to enhanced tree growth and ecosystem stability due to complementarity effects, better adaptability to changing environmental conditions and lower vulnerability to diseases, insect attacks, fire and storms[18–20]. However, to make use of the increasing knowledge about biodiversity–ecosystem functioning relationships in forest ecosystems, it would be necessary to develop methods to assess plant functional diversity efficiently over large continuous areas. Our first aim is therefore to develop such a method for a regional test area, see Fig. 1, as a base for larger scale biodiversity scoping studies.

Spatial variation in plant functional traits and diversity depend on community structure[21] and thus represent a potential signal of community assembly processes. However, plant traits and functional diversity do not only depend on community structure represented by particular species abundance distributions within a specific geographical unit, but may vary as much within species as they do between species[22]. Different species can also be redundant in terms of their functional traits, and thus not contribute to functional diversity[16, 23]. Therefore, functional diversity is best derived from a given set of traits including their intraspecific variability[24, 25]. By incorporating individual-level functional traits, functional diversity may better predict ecosystem functioning than species-level means[16].

A multitude of forest monitoring networks exist[26] as well as trait-based studies in forested ecosystems[27], fostered by standardized measurement procedures[28] and global trait databases[29]. However, these procedures usually require taxonomic information about tree individuals and indirectly assess trait variation and functional diversity combining information about species abundances and mean traits, thus ignoring variation in tree functional traits within species, which can be large even within individuals[30]. In addition, there is a global bias in the distribution of forest plots, leading to large data gaps particularly in remote areas[31]. Furthermore, trait measurements in forests are typically limited in extent and magnitude due to the complexity of destructive crown-level measurements, as well as associated georeferencing challenges and plot representativeness[32]. Consequently, continuous spatial data of traits and especially on trait diversity are still very sparse. Recent advances in remote sensing provide the opportunity to map traits and trait diversity, thus filling the existing data gaps[33–35]. Here, we use three morphological and three physiological functional traits that we assess directly, i.e. without reference to taxonomic information, to provide a spatially continuous description of functional diversity in a forest at local scale (≈925 ha), with the potential to scale up to regional and to the global level.

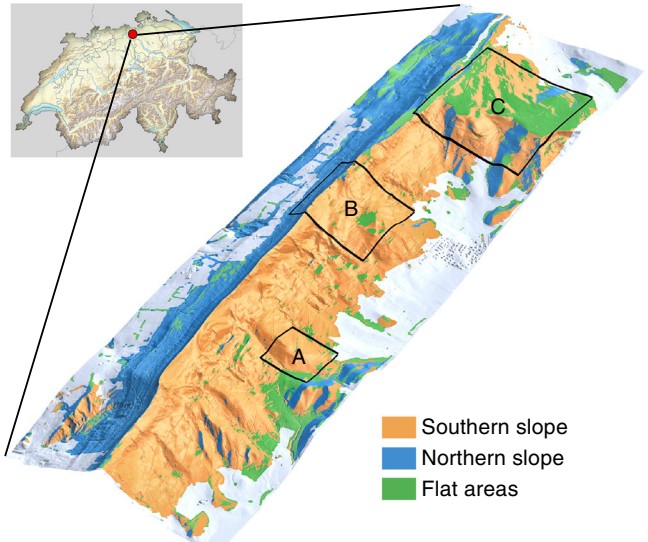

**Fig. 1** Laegern mountain temperate mixed forest site in Switzerland. The test site is located near Zurich and covers about 2 × 6 km. The mountain range is divided by a ridge running from east to west, separating the forested area in north facing (blue) and south facing (orange) slopes. Flat areas are defined with a slope <10° (green). Areas not covered by forest (agriculture, grassland, urban areas) are shown in grey

The selected morphological and physiological traits can be assessed with high-resolution airborne remote sensing methods[33, 36] and are relevant for plant and ecosystem function. Three morphological traits, namely canopy height (CH, vertical distance between canopy top and ground), plant area index (PAI, projected plant area per horizontal ground area) and foliage height diversity (FHD, measure of variation and number of canopy layers), are essential to describe canopy architecture, encompassing the horizontal and vertical structure of forests and influencing light availability, thus affecting competitive and complementary light use and ecosystem productivity[18, 37]. Three physiological traits, namely leaf chlorophyll (CHL, relative content of chlorophyll a+b per unit leaf area), leaf carotenoids (CAR, relative content of carotenoids per unit leaf area) and equivalent water thickness (EWT, leaf water content per unit leaf area), do not modify light availability but rather describe light use at the level of single leaves. The chlorophylls are functionally important pigments, since they control the amount of photosynthetically active radiation absorbed for photosynthesis[38]. Carotenoids are contributing to the chlorophylls by absorbing additional radiation for photosynthesis and protecting leaves from over-exposition to high amounts of incoming solar radiation by releasing excess energy[38]. The third, EWT, is important for plant responses to drought, which could reduce the physiological performance through decreased photosynthetic carbon assimilation and electron transport rate[39].

We use the above traits to derive measures of functional diversity separately for the morphological and leaf physiological traits. Our functional diversity measures are combining multiple traits, as is typically done for such measures[23]. We calculate three measures, related to different aspects of functional diversity—functional richness, divergence and evenness[40, 41]. Functional

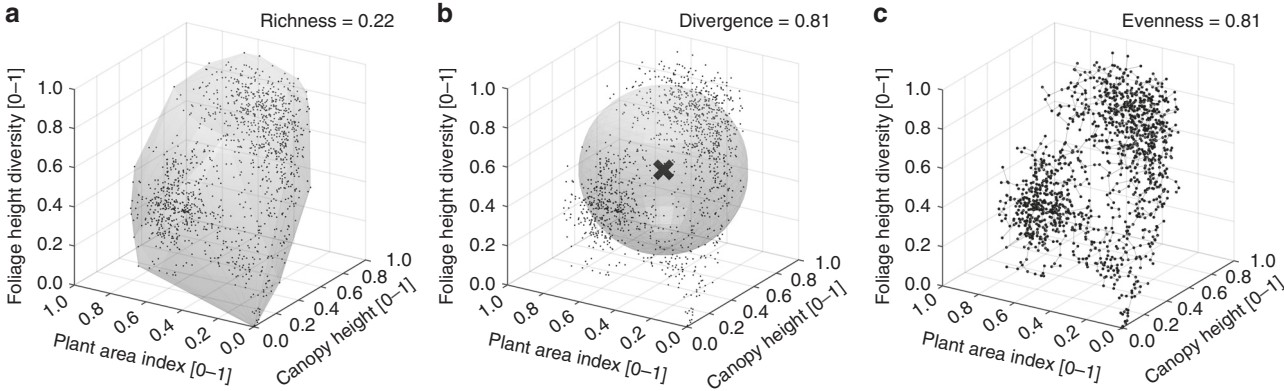

**Fig. 2** Three aspects of functional diversity based on morphological forest traits of a circular area with a radius of 120 m. The three traits are foliage height diversity, plant area index and canopy height in relative units from 0 to 1. **a** The shaded volume is functional richness, **b** the distance from the surface of the shaded sphere is functional divergence and **c** the variation of segment length in the minimum spanning tree is functional evenness

richness is calculated as the convex hull volume of the community niche[42], as illustrated in Fig. 2a for an assemblage of pixels mapped in the morphological trait space. It corresponds to the niche extent and defines the outer boundary of the occupied functional space. A disadvantage of this measure may be a strong influence by extreme values. In contrast, functional divergence and evenness describe how sample points are distributed within the community niche (Fig. 2b, c). Functional divergence is a measure of how sample points are spread with regard to the mean distance to the centre of gravity, whereas functional evenness indicates how evenly traits are distributed with regard to spacing among similar sample points in functional space. These three indices have mainly been applied to functional diversity of plants[43], where sample points represent species, with an increasing number of studies on forest ecosystems[44]. However, this concept has not yet been applied to continuously measured trait data independent of taxonomy, vegetation units or even plant individuals. Remote sensing methods offer to measure functional traits continuously and directly across large spatial extents. This has a twofold advantage: (1) there is no need to identify species, vegetation units or individuals and (2) prediction of ecosystem functions using independently established functional diversity–ecosystem functioning relationships are consistent across large scales. In contrast, recent efforts to map forest biodiversity have used forest functional classes as remotely sensed vegetation units with constant trait values assigned to these units[35].

Our second aim is to test the consistency of our method. For this, we compare the results obtained with the two independent sets of traits. Morphological diversity was found to be the main driver of forest productivity in poly- and monocultures of mature forests[45–47], whereas physiological diversity reflects different resource allocation strategies to maximize light capture and protective mechanisms and is more closely linked to species diversity[48, 49]. Since most functional traits show consistent variation along broad environmental gradients, we expect both morphological and physiological diversity to show similar patterns at larger scales. For the leaf physiological traits, we also compare the remotely sensed trait values with those directly obtained from spectroscopic measurements on single leaves. This should indicate how well the retrieval method can be scaled from the leaf to the canopy level. Furthermore, we test the general agreement of trends in trait relationships between community weighted means of the functional trait database TRY and the retrieved traits for communities composed of the 13 tree species present in our test area.

Finally, we examine scale dependency of different functional diversity measures. We demonstrate that functional diversity measures can be quantified at any desired unit area within the sampled region, limited only by the spatial resolution of the trait maps. This will allow—in future efforts—for direct and continuous mapping of functional diversity from space. Functional diversity, due to redundancy and trait plasticity, may not show the same increase with area as is typically found for species richness. Nevertheless, scale dependency of functional diversity could still lead to scale-dependent functional diversity–ecosystem functioning relationships. Such effects would be expected if ecosystem functions are not scale-dependent above a certain minimum area, which is likely the case, such as for example for productivity per area. Studies on spatial patterns and scale dependency of functional diversity are still sparse[50]. We expect functional richness to increase with scale. A strong increase at small scales would indicate high diversity within communities, which can mean higher resilience to disturbance[51], while an increase at larger scales would indicate high diversity between communities. The exact slope and shape of the relationship, however, cannot be predicted by known species–area relationships, since functional richness is influenced by trait correlations, redundancies among species and intra-specific trait variation. Even less is known about other components of functional diversity. A study based on four plant communities on the Santorini Archipelago found no relationship with area for functional divergence and evenness[52].

## Results

**Functional traits**. Figure 3 shows the spatial distribution of morphological and physiological traits, as derived from airborne laser scanning and airborne imaging spectroscopy, respectively. Blue areas in the morphological trait map are characterized by high canopy density, low canopy height and little canopy layering. When comparing with independent community data, around 83% of these areas are classified as juvenile forest with tree height below 21 m and diameter at breast height below 30 cm (Supplementary Fig. 1). The largest such area is marked as subregion A, covering ~1.4 ha, and is likely affected by disturbance caused by a winter storm. Physiologically, these patches are characterized by very high chlorophyll concentration as compared to an undisturbed, mature forest canopy.

Larger patches with a dense and closed canopy as well as high relative chlorophyll and carotenoids content are represented by pink and orange areas in the morphological and physiological

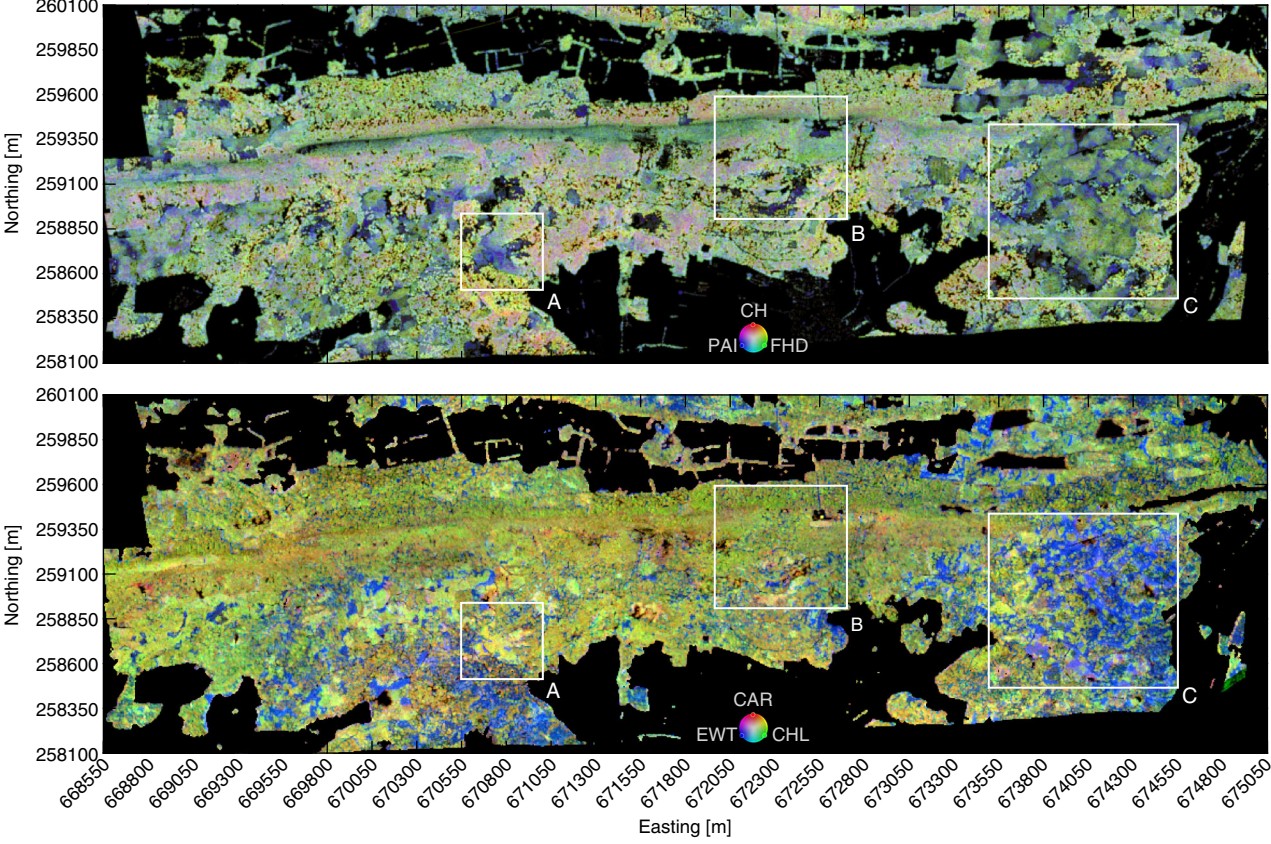

**Fig. 3** Spatial composition of morphological and physiological forest traits. RGB colour composites of morphological traits (upper panel) plotted as abundance-scaled plant area index (PAI, blue), canopy height (CH, red) and foliage height diversity (FHD, green), and physiological traits (lower panel) plotted as abundance-scaled equivalent water thickness (EWT, blue), carotenoids (CAR, red) and chlorophyll (CHL, green)

trait maps (Fig. 3). According to community data, these areas are dominated by beech trees (>50% *Fagus sylvatica*, see Supplementary Fig. 1). They appear mainly in submontane locations with shallow, but fertile alkaline soils (e.g. *Rendzina*), as well as in lower altitudes on southern slopes with deeper neutral to acidic *Podsol* and *Cambisol* soils. On the physiological trait map, blue areas with a high relative water content and low relative chlorophyll and carotenoids content are dominated by evergreen coniferous needle trees. They are more abundant in flatter areas and on southern slopes with deeper, acidic soils. These areas can further be characterized by morphological traits. A managed plantation, marked as subregion C, with 20 m tall conifers (green areas covering about 25%) can be distinguished from up to 55 m high and complexly structured canopies (yellow areas covering around 3%). The Laegern mountain is forested up to the top of the ridge, but tree height decreases to a shrub type forest with a layered but low and sparse canopy covering about 25% of the forested area. On dry and rocky habitats, sessile oak (*Quercus petraea*) and beech (*Fagus sylvatica*) are the dominating tree species.

PAI, CH and FHD have a mean and standard deviation of $0.46 \pm 0.21$, $0.49 \pm 0.17$ and $0.59 \pm 0.20$, respectively, when normalized between 0 and 1 (Supplementary Fig. 2a). CH and FHD are correlated with $r^2 = 0.70$, CH and PAI with $r^2 = 0.31$, and FHD and PAI with $r^2 = 0.35$ (Supplementary Fig. 3). Figure 4 shows median and standard deviation of the functional traits along altitudinal belts. 5.5%, 10.5% and 5.6% of the variance in CH, PAI and FHD can be explained by soil and topography (Supplementary Fig. 4). Soil variables alone explain 1.4%, 8.2% and 4.1% of

the variance, respectively. CHL, CAR and EWT have a mean and standard deviation of $0.58 \pm 0.18$, $0.50 \pm 0.21$ and $0.39 \pm 0.18$, respectively, when normalized between 0 and 1 (Supplementary Fig. 2b). CHL and CAR are correlated with $r^2 = 0.57$, CHL and EWT with $r^2 = 0.004$, and CAR and EWT with $r^2 = 0.08$ (Supplementary Fig. 3). 11.9%, 20.3% and 34.8% of the variance in CHL, CAR and EWT can be explained by soil and topography. Soil variables alone explain 9.9%, 14.1% and 27.5% of the variance. Radiation is correlated with soil and topography ($r^2 = 0.56$) and therefore only explains an additional 0.1–0.5% of variance of the functional traits (Supplementary Figs. 4 and 5).

Estimated physiological trait ranges based on imaging spectroscopy correspond with modelled ranges based on leaf optical properties measured in the field (Supplementary Fig. 6). General trends of community-weighted mean trait values agree with the functional trait database TRY (Supplementary Fig. 7). Although TRY is not suitable for assessing intra-specific trait variation or trait plasticity, we find a positive relationship to remotely sensed trait estimates of chlorophyll ($r^2 = 0.36$) and EWT ($r^2 = 0.48$). Simulations using lab measurements of traits and leaf optical properties in a 3D forest model show that spectral indices can be applied at the canopy level, if high quality imaging spectroscopy data with little influence of shadows are available (Supplementary Figs. 8 and 9). Canopy reflectance-based estimates of chlorophyll and carotenoids ($<15 \, \mu g/cm^2$) correlate with traits measured in the laboratory ($r^2 = 0.86$, $r^2 = 0.74$). The weakest correlation between lab measured traits and estimates from canopy spectra could be observed for EWT ($r^2 = 0.51$, Supplementary Fig. 9), since water absorption was measured in the near infrared where

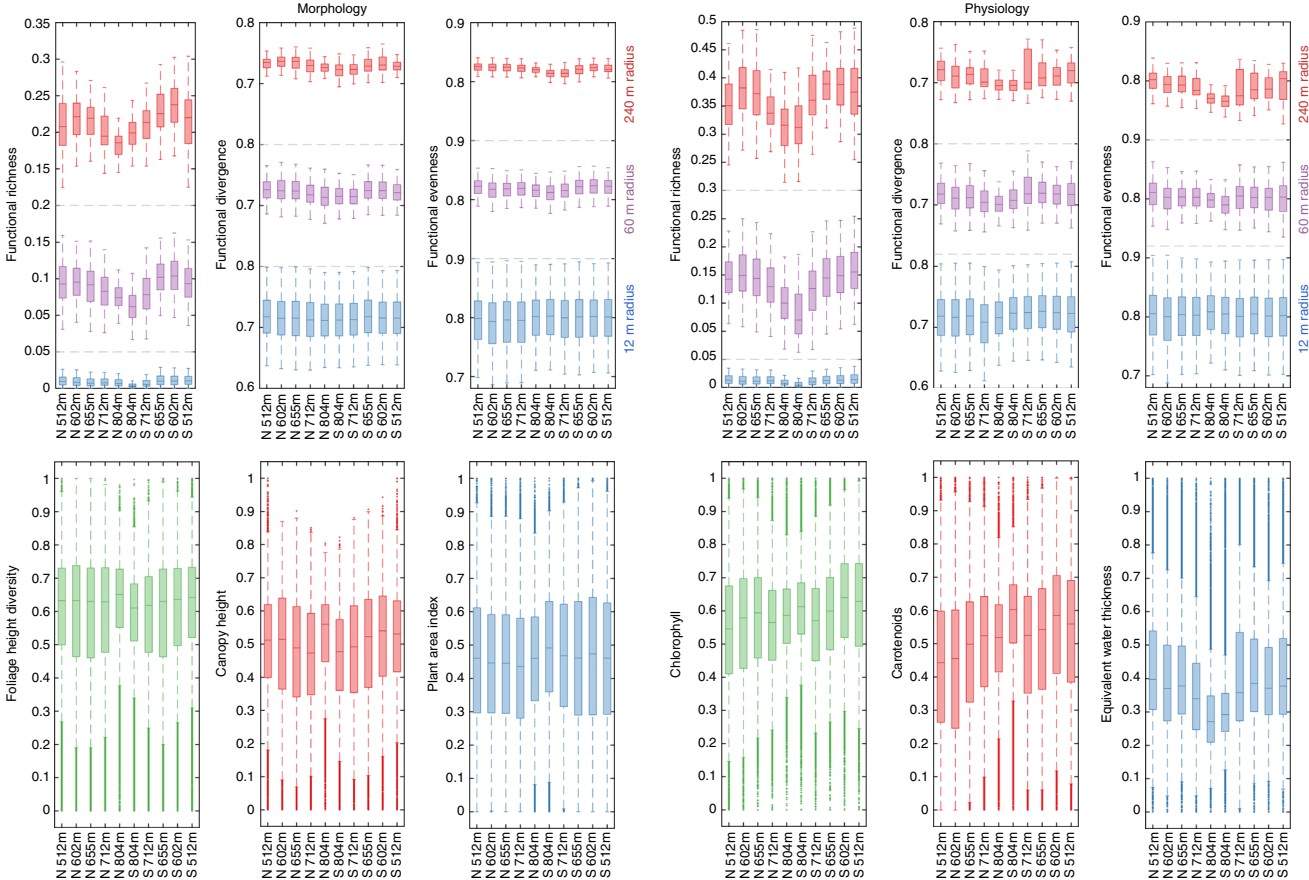

**Fig. 4** Boxplots of functional diversity indices and traits grouped by altitudinal belts. Functional richness, divergence and evenness (top panel) are shown for three spatial scales at 12 m (blue), 60 m (purple) and 240 m (red) radius. Underlying functional trait values are displayed below, with morphological traits on the left and physiological traits on the right side. Boxes show the median and ±1 standard deviation and whiskers mark ±2 standard deviations. Altitude values on the *x*-axis of the boxplots indicate the middle of the altitudinal belt for the north (N) and the south (S) side of the mountain ridge

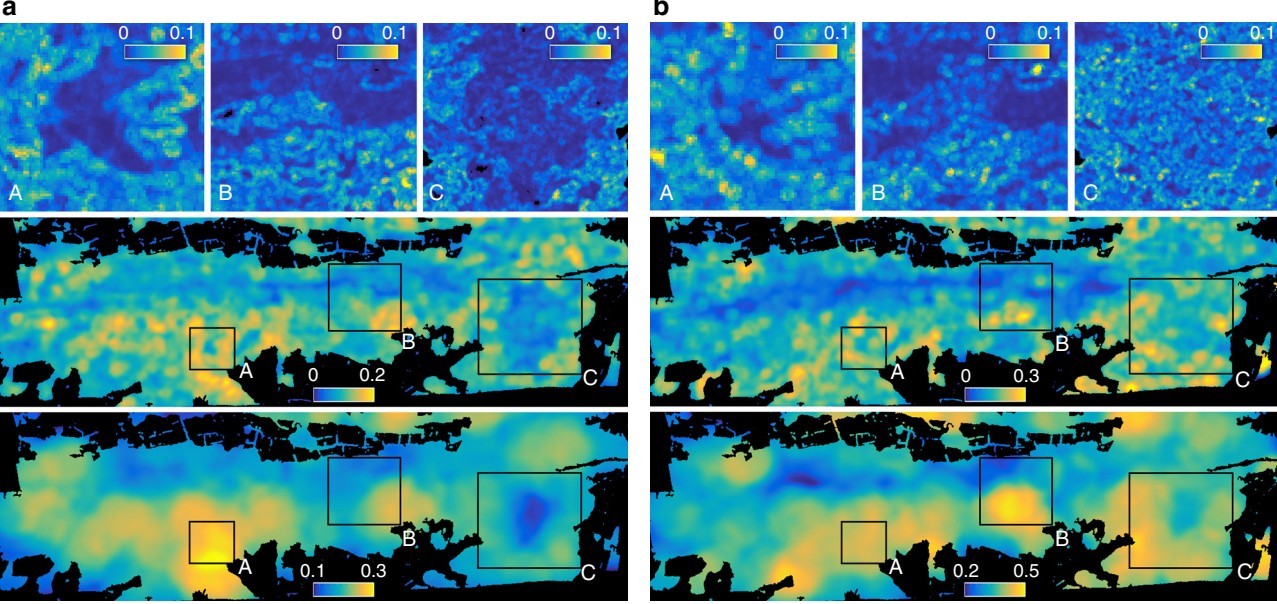

**Fig. 5** Spatial patterns of morphological and physiological richness at three different scales. Functional richness was computed at 12 m (top), 60 m (middle) and 240 m (bottom) radius based on **a** morphological traits and **b** physiological traits. At 12 m radius (top panels), subregions A, B and C are plotted only. The colour is scaled from the lowest (dark blue) to the highest (yellow) richness value with a maximum possible range from 0 to 1

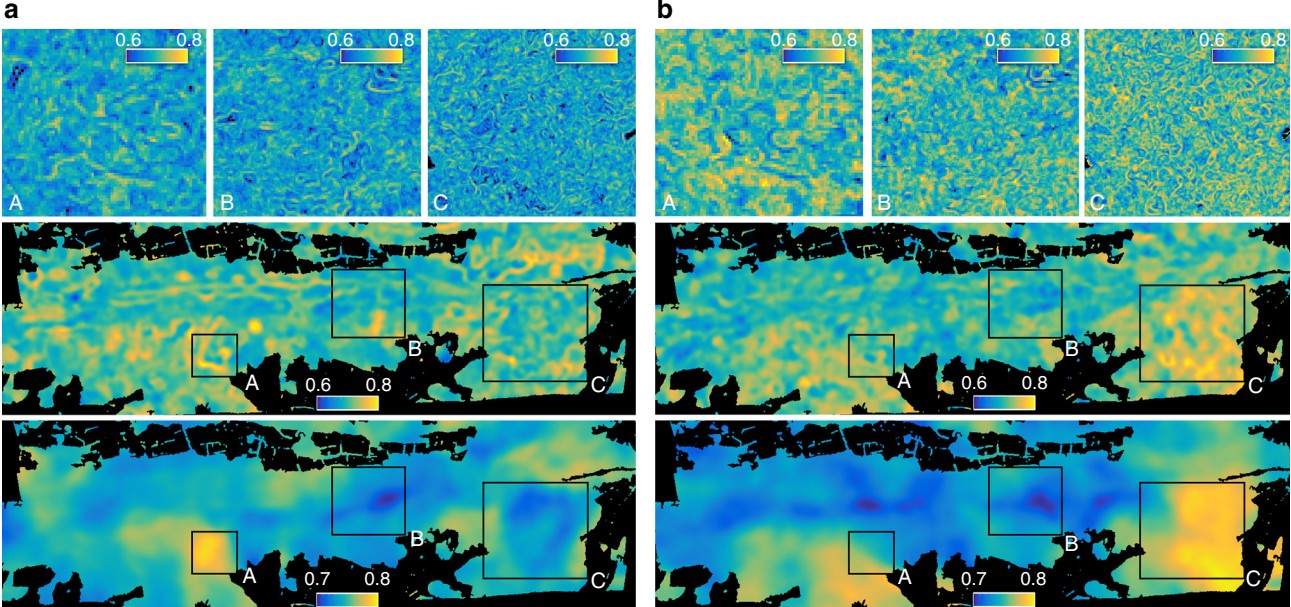

**Fig. 6** Spatial patterns of morphological and physiological divergence at three different scales. Functional divergence was computed at 12 m (top), 60 m (middle) and 240 m (bottom) radius based on **a** morphological traits and **b** physiological traits. At 12 m radius (top panels), subregions A, B and C are plotted only. The colour is scaled from the lowest (dark blue) to the highest (yellow) divergence value with a maximum possible range from 0 to 1

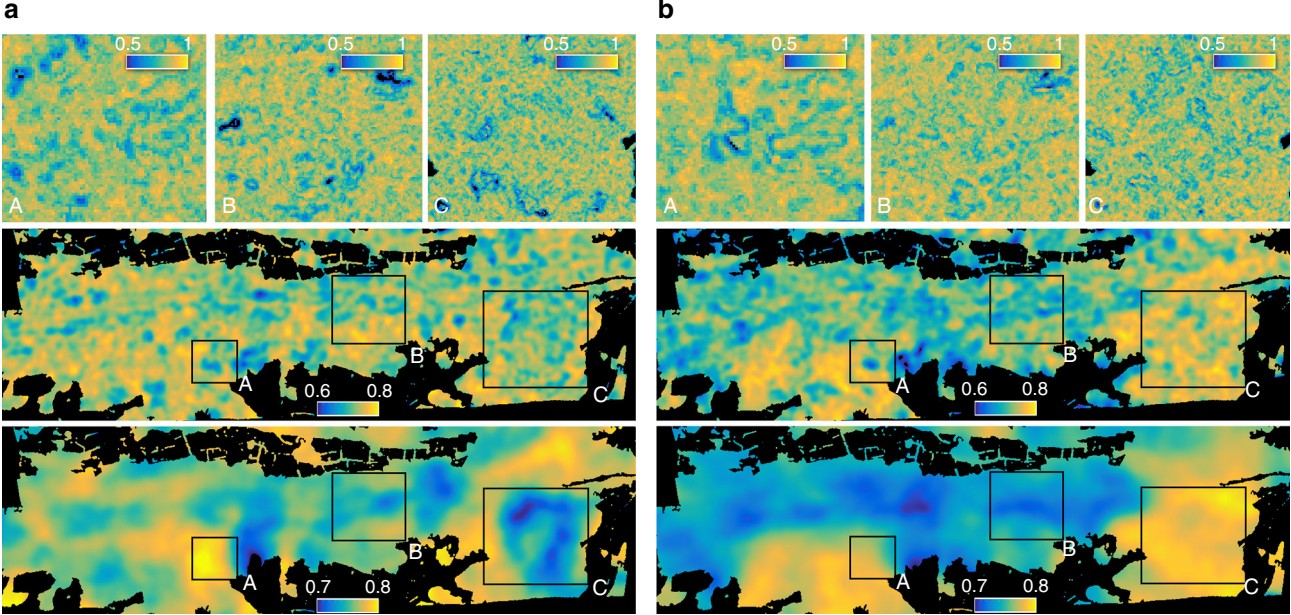

**Fig. 7** Spatial patterns of morphological and physiological evenness at three different scales. Functional evenness was computed at 12 m (top), 60 m (middle) and 240 m (bottom) radius based on **a** morphological traits and **b** physiological traits. At 12 m radius (top panels), subregions A, B and C are plotted only. The colour is scaled from the lowest (dark blue) to the highest (yellow) evenness value with a maximum possible range from 0 to 1

scaling from leaf to canopy level is hampered by multiple scattering effects.

**Functional diversity**. Maps of functional richness, divergence and evenness are shown in Figs. 5–7. Patterns of morphological and physiological richness exhibit strongest correlation at medium scale between 60 and 240 m radius. The correlation coefficient ($r$) is 0.37, 0.44 and 0.40 at 12, 60 and 240 m radius, respectively. Differences among northern, southern and flat areas

are significant for both morphological (DF = 2, $F$ = 5.8, $p$ < 0.01) and physiological richness (DF = 2, $F$ = 9.1, $p$ < 0.01) based on a generalized linear model and an ANOVA test. Figure 4 shows a consistent decrease of functional richness towards the mountain ridge for morphological and physiological richness. Soil and topography together explain 24.2% and 40.1% of variance in morphological and physiological richness, whereas 19.6% and 34.6% of variance is explained by soil alone and 15.3% and 37.9% by topography alone (Supplementary Fig. 4). For morphological richness, altitude (DF = 1, $F$ = 48.4, $p$ < 0.001) and curvature (DF

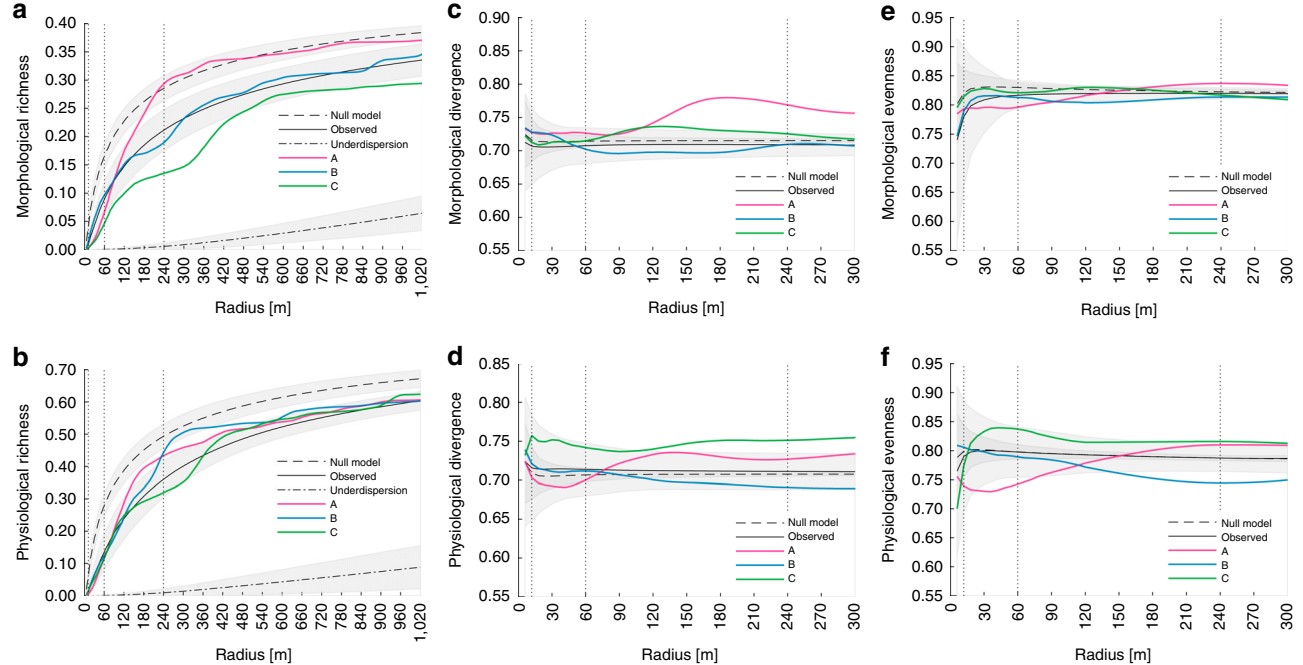

**Fig. 8** Scale dependency of the three functional diversity measures for morphological and physiological traits. Functional **a**, **b** richness, **c**, **d** divergence and **e**, **f** evenness are displayed as a function of radius (diversity–area) for **a**, **c**, **e** morphological and **b**, **d**, **f** physiological traits. Null model corresponds to randomly distributed traits as dashed line. Observed corresponds to remotely sensed data as solid line. Underdispersion assuming direct neighbours corresponds to dashed–dotted line. Curves A, B C are stemming from a single pixel in the centre of subregions as in Fig. 5. Vertical dotted lines correspond to radii as in Fig. 5

= 2, $F = 3.8$, $p < 0.05$) explain most of the variance. Physiological richness is more strongly linked to slope (DF = 1, $F = 121.5$, $p < 0.001$), being steepest on the south side of the ridge and indirectly linked to radiation, followed by altitude (DF = 1, $F = 20.5$, $p < 0.001$). With slope explaining most of the variance, aspect is not significant any more (Supplementary Table 1).

The correlation ($r$) between patterns of morphological and physiological divergence is 0.36, 0.13 and 0.21 at 12, 60 and 240 m radius, respectively. Divergence remains in a relatively small range, leading to small relative differences between high and low diversity areas. Only altitude is significantly related to morphological divergence (DF = 1, $F = 8.4$, $p < 0.01$) based on a generalized linear model and an ANOVA test, whereas variance in physiological divergence is mainly explained by slope (DF = 1, $F = 23.4$, $p < 0.001$). Soil and topography together explain only 7.7% and 17.4% of total variance, with soil being the more important factor. Functional evenness patterns of morphological and physiological traits strongly correlate at small scales, for example with a correlation coefficient ($r$) of 0.54 at 12 m radius. The correlation decreases towards 0.19 and 0.23 at 60 and 240 m radius, respectively. Evenness is slightly higher on southern than on northern slopes and flat areas, but the deviation from the average is below 2% for morphological and below 3% for physiological traits. Morphological and physiological evenness vary mainly with altitude (DF = 1, $F = 14.0$, $p < 0.001$) and slope (DF = 1, $F = 14.8$, $p < 0.001$) respectively. Similar to divergence, soil and topography explain 10.7% and 12.1% of variance, respectively.

Figure 8 shows how functional richness of morphological and physiological traits change as a function of spatial scale (see Supplementary Fig. 10 for mean and standard deviations of all pixels in the subregions). Fitting a power-law function to the observed mean functional richness–area relationship results in a slope of 0.195 and 0.213 for morphological and physiological richness, respectively (Supplementary Fig. 11). This is close to

previously reported slopes for species richness–area relationships of 0.161 and 0.177 for the biome 'temperate broadleaf and mixed forest' and the land-cover class 'temperate mixed forest'[53]. The increase of functional richness with the logarithm of the area is linear for areas above 1 ha. Therefore, a logarithmic function fits the mean observed values better than the power-law, although both $r^2$-values are very high ($r^2 > 0.9$). In contrast, mean functional divergence and evenness are scale invariant. They remain stable with changing extent (<1.3% change between radius 60 and 240 m) and do not vary between different trait values and distributions (Fig. 8c–f). The main difference between observed and random spatial distribution of traits is the magnitude of the variance.

A modelled random spatial distribution of functional traits (Fig. 8, null model), preserving the relationship among the three traits of a pixel, leads to a relatively high functional richness. Especially at smallest scales below 50 m radius, richness following a random distribution is one to five times higher than based on the observed distribution of traits. Null model richness is 35% and 37% higher at 240 m radius, decreasing to 15% and 11% at the largest scales for morphological and physiological traits respectively. A simulated distribution of traits following the assumption of under-dispersion, where trees being close in functional space are assumed to be close in geographic space, leads to a very low functional richness at all scales (<19% (a) and <15% (b) of observed values).

## Discussion

Various measures of functional diversity, with different advantages and disadvantages[54, 55], have the common aim to map species in functional trait space using mean trait values and weighted species abundances[40, 41]. With our new method we create continuous maps of functional diversity without a need to identify species or individuals, since inter- as well as intra-specific

variability is inherent to remotely sensed functional traits. Especially in relatively species-poor temperate forests, such as the one studied here, functional diversity might be strongly underestimated when ignoring intra-specific variability[16, 56]. Our method avoids this pitfall, because it is fully continuous in space and only depends on resolution, it can thus even be applied below the individual level. Within-individual variation, for example in leaf traits, is common in plants and can reflect different light competition or leaf ages[30]. With evolving sensor technologies and miniaturization, higher spectral and spatial resolution of remotely sensed data will allow to study within-individual tree functional diversity.

The resulting spatial distribution of morphological and physiological diversity generally agree with regard to the spatial patterns, especially for functional richness. This is related to the environmental gradient on the mountain in the observed test area and the coinciding reduced trait variability towards the ridge (Fig. 4). The mountain ridge is the most prominent landscape feature of our study area with shallow and rocky soil, steep slopes and high incoming radiation on the south side of the ridge (Supplementary Fig. 5). We can therefore show that both morphological and physiological diversity change consistently with topography and soil. In this case, the abiotic conditions at the ridge might act as an environmental filter, only allowing trees with particular functional traits to exist. This is important because functional richness represents the total extent of the community niche. The lower functional richness at higher elevation with dry, rocky and shallow soil suggests a smaller range of resource availability. Thus smaller biotope space constrained the community niche in this area, which as a consequence may reduce the performance of the present plant community[57] and its adaptability to changing environmental conditions[40]. Therefore, we would expect the forest communities on the ridge to have lower ecosystem functioning and stability.

Besides similarities in the spatial distribution of functional diversity following broad environmental gradients, there are also expected differences between morphological and physiological diversity. These differences are more pronounced for functional divergence and evenness than for functional richness. On the one hand, physiological divergence is mainly driven by differences between tree functional groups (needle, broadleaf), because they have different leaf structure and composition of pigments and compounds related to different resource allocation strategies and are therefore clearly divergent in their biochemical characteristics. In areas with mixtures of broadleaf and needle trees, as for example in subregion C or generally in lower altitudes, productivity might be increased because the resource use is partitioned among the different functional groups leading to lower resource competition[40]. At the same time, functional evenness is higher too, indicating that the niche is filled evenly and available resources can potentially be fully exploited. In higher altitudes where the trait range is reduced, lower divergence and evenness could mean that there is a stronger competition for resources (nutrients, water) and that some of the resources might be unused, leading to lower productivity and stability of the community.

Morphological diversity, on the other hand, is more strongly linked to the different stages of forest development (e.g. due to disturbance) and management. For example, subregion A shows high morphological diversity at larger scales because there is a juvenile forest patch in the centre surrounded by structurally different mature trees. In contrast, morphological richness, divergence and evenness are low in the managed forest in subregion C due to equal canopy height and structure. This may result in lower productivity due to a lower efficiency in light capture, although higher physiological diversity could indicate better resource use partitioning among functional groups. The

strong link to the development stage is clearly reflected in the morphological traits themselves. Differences in functional traits between juvenile and mature forest communities can be explained by changing physiology and morphology with tree age, ranging from densely and fast growing highly productive juvenile to mature trees, being characterized by lower growth rate, similar height, smaller leaves and greater leaf thickness and longevity[58]. Since the occurrence of patches of juvenile forest is mainly driven by disturbance and forest management, there is no clear altitudinal gradient in functional traits.

In contrast, physiological traits are linked more closely to topographic and soil variables. Equivalent water thickness in particular shows the strongest altitudinal gradient, because there is a gradient in soils and steepness leading to lower potential water availability towards the top of the ridge. Furthermore, needle trees mainly occurring in lower altitudes show higher EWT and lower relative chlorophyll and carotenoids content compared to broadleaf trees. This is in accordance with values from the TRY database (Supplementary Fig. 7) and a study conducted at three sites in Switzerland, reporting higher water and lower nitrogen content, being closely linked to chlorophyll content[59]. In general, our remotely sensed functional traits are consistent with independent in situ knowledge of the forests in the study region. We could show that functional traits are mapped in the correct range and that our measurement values are compatible with values derived from optical and functional trait databases. To map functional diversity, relative trait values can be used but they need to be measured consistently over space. The proposed remote sensing method has the advantage that it is based on continuous and consistent large-scale measurements without bias due to subjective interpretation or differences in measurement techniques or protocols, which can occur when traits are measured over large areas in the field.

Given the continuous nature of the remotely sensed functional trait maps, we were able to study functional diversity at multiple scales and to develop a highly resolved scaling relationship. The relationship of functional richness and area should be related to the species–area relationship, which is one of the most studied ecological patterns due to its relevance for predicting biodiversity patterns and species extinction rates[53]. Typically, the power-law is used to model species–area relationships resulting in a linear relationship on the log–log scale. Our results are generally consistent between morphological and physiological richness. Furthermore, the slope of the relationship on the log–log scale is very similar to large-scale species models for temperate mixed forests[53]. However, we also found deviations of the relationship from the power-law, as was also reported by Pereira and colleagues for smaller spatial scales[60]. Increased within-community diversity when considering intra-specific variability might explain the steeper slope at small scales, whereas species might be redundant with regard to their functional traits at large scales, leading to a flattening of the log–log relationship. Therefore, we found that a logarithmic function could better predict functional richness than did the power-law.

Deviations from the average can be observed locally, when looking at particular subregions within the test area. Exemplary for a steep transition from low to high functional richness with increasing area is subregion A. Juvenile trees that grow in a disturbed area result in low within and high between community diversity. In this case, underdispersion at local scale might not only be driven by abiotic conditions (e.g. environmental filtering) or anthropogenic influence but also by competitive exclusion[61]. Beech trees might have been planted in disturbed areas or favoured by environmental conditions, or both, but at the same time only the fastest growing beech trees with similar functional traits might have survived and occupied the new space. When

competing for light, a competitive ability difference leads to the elimination of individuals that grow slowly and are therefore too short to gather enough light[61]. According to Siefert[62], local under-dispersion leads to locally decreased functional divergence and increased divergence between environmental patches. This is in agreement with what we observed in subregion A (Fig. 8).

By comparing functional richness–area relationships of observed with randomly distributed traits, we found trait convergence to be predominant in our forest. However, a general link between community structure and underlying assembly processes can not easily be established, because many processes can lead to trait divergence or convergence, including anthropogenic factors due to certain management strategies. Opposing processes can balance each other and not be disentangled any more[44, 63, 64]. The latter might be the case when looking at the average signal of functional divergence and evenness, which is scale invariant and almost similar to the null model. This, however, does not mean that there is no spatial variation of these two aspects of diversity at all. To study the scale dependency of biodiversity, it is therefore crucial to not only focus on general relationships but also on spatially continuous diversity patterns at different scales.

In conclusion, combined airborne imaging spectroscopy and laser scanning allow for mapping functional diversity continuously across large areas of forest using a trait-based, pixel-level approach. We evaluated the diversity of six key traits at a variety of spatial scales and were able to validate these measurements against in situ data, as well as to assess community structure across an entire landscape. By concentrating on functional traits at a continuous spatial resolution without reference to species identities or individuals, we were able to include intra-specific variability, which is crucial to assess functional diversity of temperate forests and often neglected when functional diversity is indirectly calculated from taxonomic data. Future studies can advance the integration of remotely sensed functional data with databases of plant functional traits, environmental and ecosystem data, and dynamic vegetation models to increase our understanding of the mechanistic linkages between functional diversity and ecosystem function.

To map functional diversity from space and predict global patterns of ecosystem functioning, our method could also be applied to satellite measurements, even though at lower spatial resolution. To test the scalability of our approach we suggest looking at changing extent and grain in a combined fashion. Supplementary Figure 12 indicates how well richness patterns correlate at a given neighbourhood radius when changing grain as pixel size. For example, satellite data at 30 m spatial resolution might be able to capture richness patterns at 200 m radius with a correlation coefficient of 0.7–0.8. This paves the way for possible large-scale applications, but further research is needed to quantify how much small-scale variability would be lost when pixel size is increased, and how this would affect diversity–productivity relationships.

## Methods

**Study area.** The study area is a temperate mixed forest at the Laegern mountain in Switzerland (47° 28′43.0 N, 8° 21′53.2 E). The Laegern is characterized by a mountain ridge spanning in east–west direction with an altitudinal gradient of 450–860 m above sea level (Fig. 1). The extent of the study area is ∼2 km × 6 km. In December 1999, the Laegern mountain was affected by a winter storm. The western part of the temperate forest was severely hit, resulting in disturbance areas filling in with beech trees as new stands are initiated. Since forest clear cuts are limited to a maximum area of 0.5 ha, larger patches of juvenile trees likely exist due to the storm. In 2010, the juvenile trees were 10–15 m high and growing in dense patches with a growth rate of around one metre per year[65]. The mainly closed canopy consists of a total of 13 species and seven canopy structure types, from single- to multi-layered canopies[66]. Roughly 70% of the total forested area is covered by deciduous broadleaf trees, whereas the remaining 30% of the area is covered by evergreen coniferous trees (forest inventory data). The dominating deciduous

species are common beech (*Fagus sylvatica*), European ash (*Fraxinus excelsior*) and sycamore maple (*Acer pseudoplatanus*). The dominating coniferous species are Norway spruce (*Picea abies*) and silver fir (*Abies alba*). Most of the conifers at Laegern were introduced anthropogenically. Naturally, the whole Laegern forest would be dominated by different hilly to submontane beech communities with few scattered coniferous needle trees. There are mature trees up to 165 years of age, 150 cm of diameter and canopies up to 55 m of height. The study area comprises a reference site for forest ecosystem research with an extensive set of ground measurements[36, 66].

**Airborne remote sensing data.** The data of the Laegern study area was acquired in 2010 using airborne laser scanning based on the principle of light detection and ranging (LiDAR) and airborne imaging spectroscopy. The LiDAR acquisition was flown on 1 August 2010 using a helicopter-based scanner system with a rotating mirror (RIEGL LMS-Q680i, scan angle ±15°). The campaign was flown under leaf-on conditions with a nominal height of 500 m above ground, resulting in a footprint size of 0.25 m and an average point density of 40 pts/m². The 3D point cloud was extracted from the full waveforms of individual laser pulses using Gaussian decomposition. The LiDAR data was registered to the Swiss national grid CH1903+ with a positional accuracy of <0.15 m in vertical and <0.5 m in horizontal direction.

Imaging spectroscopy acquisitions were flown on 26 June and 29 June 2010 under clear sky conditions using the APEX imaging spectrometer[34]. The study area was covered with three flight lines on each of the acquisition dates. The average flight altitude was 4,500 m a.s.l. resulting in an average ground pixel size of 2 m. APEX measured at-sensor radiances in 316 spectral bands ranging from 372 nm to 2,540 nm. APEX data were processed to hemispherical-conical reflectance factors in the APEX Processing and Archiving Facility[67]. Processing started with the raw instrument data, which was split into image, dark current and housekeeping data, thus forming level 0. Level 1 (L1) calibrated radiances were obtained by inverting the instrument model, applying coefficients established during calibration and characterization at the APEX Calibration Home Base[68]. The position and orientation of each pixel in 3D space was based on automatic geocoding in PARGE v3.2[69], using the swissALTI3D digital terrain model. L1 data were then converted to HCRF by employing ATCOR4 v7.0 in the smile aware mode. This essentially accounts for the spectral response function of each individual pixels of the spectrometer to reduce biases due to spectral shifts[34].

**Environmental data.** Stand polygons of Kanton Aargau and Zurich include forest stand information on development stage, the percentage coverage of the six most dominant species, and the percentage coverage of deciduous broadleaf and coniferous needle trees. The data from Kanton Aargau was provided by Aargauisches Geografisches Informationssystem (AGIS), Departement Bau, Verkehr und Umwelt, Abteilung Wald (last updated on 27 February 2015). The data from Kanton Zurich was provided by Geographisches Informationssystem (GIS-ZH), Amt für Landschaft und Natur, Abteilung Wald (last updated on 16 September 2015). Soil data corresponds to Bodenkarte Baden (Landeskarte der Schweiz 1:25′000, Blatt 1070), provided by Eidgenössische Forschungsanstalt für Agrarökologie und Landbau (FAL).

Topographic variables (altitude, slope, aspect, curvature) were calculated based on the digital terrain model derived from a LiDAR acquisition on 10 April under leaf-off conditions. The campaign was flown with a nominal height of 500 m above ground, resulting in a footprint size of 0.25 m and an average point density of 20 pts/m². Radiation was simulated as incoming photosynthetically active radiation at the top of canopy (see Supplementary Note 1 for details). Supplementary Fig. 13 shows a comparison between simulated and measured radiation at the fluxtower in the Laegern forest.

**Field data.** At the Laegern reference site, field survey was conducted on an area of ∼5.5 ha to map the exact ground location and taxonomic identity of all dominant and co-dominant trees (1,307 trees with dbh >20 cm). The positions measured on the ground were linked to a detailed crown map derived from high-resolution drone images. Leaf optical properties of sunlit leaves were measured for ten *Acer pseudoplatanus*, *Fraxinus excelsior*, *Fagus sylvatica*, *Ulmus glabra* and *Tilia platyphyllos* trees in June 2009 and for 50 *Fagus sylvatica* trees in July 2016. For the 50 trees, SPAD measurements were taken of the same leaves. Leaf optical properties and lab measured traits (chlorophyll, carotenoids, EWT) of 168 *Acer pseudoplatanus* trees were used from the ANGERS spectral database.

**Functional traits.** Functional traits were measured and mapped using state-of-the-art airborne remote sensing methods. A set of three morphological and three physiological traits was selected and mapped based on airborne laser scanning and imaging spectroscopy data respectively. The whole work-flow from remote sensing data to functional diversity measures is illustrated in Supplementary Fig. 14.

We selected CH, PAI and FHD as the three main morphological traits, being of high ecological relevance and measurable using airborne laser scanning methods. CH was measured as the distance between the highest laser return from the canopy and the corresponding ground point following Schneider et al.[36]. PAI was retrieved as the projected surface area of plant material per unit ground area. This includes

woody as well as foliar material, since laser returns from twigs or leaves can not be distinguished. PAI was derived from the LiDAR point cloud data on a $2 \times 2\,m$ grid[36, 65]. FHD is a measure of canopy layering and has been recognized as a major functional trait for characterizing biodiversity of a variety of species and habitats[70]. FHD was calculated by applying the Shannon–Wiener diversity index on vertical PAI profiles as described by MacArthur and MacArthur[71]:

$$\mathrm{FHD} = -\sum_i p_i \cdot \log_e p_i, \qquad (1)$$

where $p_i$ is the proportion of the total foliage which lies in the $i$th canopy layer. FHD is a combined measure of how different the layers are with respect to layer density (PAI) and how many layers there are in total. Therefore, a certain correlation to CH can be expected, since the maximum possible number of layers is given by the canopy depth in conjunction with the vertical resolution of the laser system. The three morphological traits were normalized to values between 0 and 1 and resampled to $6 \times 6\,m$ spatial resolution using bilinear interpolation, approximating the average basal crown area of the Laegern forest.

Gitelson et al.[72] developed a band specific model to derive CHL and CAR from imaging spectroscopy data in relative units. It has been applied to a wide range of ecosystems, from crops to grasslands and forests[38, 73]. To derive CHL and CAR using the three-band model[72], the following band combinations were used:

$$\mathrm{CHL} = \left( \frac{1}{R_{540-560}} - \frac{1}{R_{760-800}} \right) \cdot R_{760-800}, \qquad (2)$$

$$\mathrm{CAR} = \left( \frac{1}{R_{510-520}} - \frac{1}{R_{690-710}} \right) \cdot R_{760-800}, \qquad (3)$$

where $R_{i-j}$ is the mean reflectance in the spectral range of $i$ to $j$ nanometre. The model includes anthocyanins as a third pigment[72]. We decided not to include it in our study, since anthocyanins can mainly be observed during leaf development or leaf senescence[38]. Concentrations are generally low during the summer months and are difficult to detect, since the absorption features are strongly overlapping with chlorophyll and carotenoids absorption.

As a third physiological trait, we included EWT. We estimated relative EWT with a simple ratio water content index based on Underwood et al.[74]:

$$\mathrm{EWT} = 1 - \frac{R_{1,193}}{R_{1,126}}, \qquad (4)$$

where $R_i$ is the reflectance at $i$ nanometre.

To reduce the effects of shadows in the traits retrieval, we combined two airborne imaging spectroscopy acquisitions flown at different times of the day and aggregated $3 \times 3$ pixels to $6 \times 6\,m$ resolution trait data by averaging the three brightest pixels. To fuse the flight lines, we performed an additional geometrical co-registration using scale-invariant feature transform and random sample consensus algorithms of the VLFeat package (VLFeat, sift_mosaic, Matlab). Finally, we normalized to values between 0 and 1.

Estimating physiological forest traits from airborne observations is a challenging task due to the difficulty of linking leaf and canopy level biochemistry. Airborne imaging spectroscopy measures a spatially integrated signal of the sunlit upper canopy of the forest. The mapping of functional diversity relies on relative trait values being derived from these consistent radiometric measurements. The relationship of relative trait values and their physical counterparts can be demonstrated by parametrizing the radiometric simulation of selected species with field data and generic data from two functional trait databases. The ranges of physiological traits were compared with modelled trait ranges based on the leaf optical properties measured in the field in July 2009 (Supplementary Fig. 6). The same modelling framework as in ref. [36] was used to simulate canopy reflectance spectra and subsequently derive physiological traits. Constant optical properties for broadleaf and needle trees were expected to result in a narrower trait range due to the lack of intra- and inter-specific trait variability within functional groups. For further details on the modelling approach, see Supplementary Note 2.

Field data of the 5.5 ha area at Laegern was used to calculate community-weighted mean chlorophyll and EWT. Species abundances and mean traits were calculated per $30 \times 30\,m$ plot. Remotely sensed mean trait values were then compared to community-weighted means of the functional trait database TRY[29], based on the plot-level species abundances and species-level trait values from TRY (Supplementary Fig. 7). There were not enough measurements in the TRY database to calculate community-weighted means of carotenoids.

To illustrate the scalability of the spectral indices from the leaf to the canopy level, we used the field data to simulate canopy reflectances for the 518 *Fagus sylvatica* and the 168 *Acer pseudoplatanus* trees on the 5.5 ha area. We used the leaf optical properties of 50 *Fagus sylvatica* trees measured in July 2016, and randomly distributed them over the 518 *Fagus sylvatica* trees according to field survey. Chlorophyll values were then derived from the reflectance spectra at leaf and canopy level, to be compared to the SPAD measurements of the same leaves (Supplementary Fig. 8). Additionally, we simulated canopy spectra for the 168 *Acer pseudoplatanus* trees with leaf optical properties of the ANGERS database. Lab measurements of chlorophyll, carotenoids and EWT from the database were then

compared to traits estimated using spectral indices at leaf and canopy level (Supplementary Fig. 9). Since we did not expect very high carotenoids concentrations at Laegern in summer, we fitted a second linear regression in Supplementary Fig. 9c, d for values below $15\,\mu g/m^2$. For further details on the modelling approach, see Supplementary Note 2.

For mapping in Fig. 3, we used red, green and blue (RGB) colour composites of the three normalized morphological and physiological traits respectively. We define blue areas in the morphological trait map as values of CH<0.5, FHD<0.5 and PAI>0.5, pink areas as CH>0.5, FHD>0.5 and PAI>0.5, and green areas as CH<0.5, FHD>0.3, PAI<0.5. A small area appearing yellow is defined by CH>0.7, FHD>0.7 and PAI<0.6. In the physiological trait map, we define blue areas as values of CHL<0.5, CAR<0.5 and EWT>0.5, bright green areas as CHL>0.8, CAR>0.5 and EWT<0.5, and green areas as CHL>0.5, CAR<0.5 and EWT<0.5. Orange areas are characterized by CHL<0.7, CAR>0.7 and EWT<0.5.

The forested area was determined based on CH. To derive the forest mask, we first applied a threshold of 10 m CH to select the mature forest pixels and remove possible agricultural fields. We then filled the gaps within the forest to include juvenile forest patches again. Finally, a threshold of 4 m CH was applied to remove gaps and understorey vegetation. We defined a tree to be four or more metres high, as was done in Schneider et al.[36] to separate understorey and the canopy.

**Functional diversity.** Having tens to hundreds of thousands of pixels to map is computationally demanding, guiding our choice of index. As a consequence, we selected functional richness, divergence and evenness being computationally manageable and relatively easy to interpret, since different aspects of functional diversity are covered by separate indices. The indices for functional richness, divergence and evenness were calculated based on the remote sensing derived physiological and morphological traits. We mapped pixels within a certain radial neighbourhood in the functional trait space, using a moving window approach with varying neighbourhoods to cover the whole study area. Figure 2 shows an example of functional richness, evenness and divergence calculated based on pixels in a radius of 120 m mapped in trait space. Abundance weighting is not needed since every pixel represents a set of trait measurements, not averaged by communities or species. With continuous area-based data, however, a single pixel does not necessarily cover an individual crown. Contributions of more than one individual or species to the functional traits of a singular pixel is possible and therefore represents no direct link to species. Detailed information on the three indices and pixel based application is given in the following paragraphs.

Functional richness is a measure of niche extent, where niche is the functional space occupied by a species, community or assemblage of trees. It was calculated by mapping pixels of a certain neighbourhood in functional space, whose axes are defined by the functional traits. Richness was then calculated as the convex hull volume of the mapped pixels (convhull, Matlab). Supplementary Figure 15 illustrates an artificial example of an increasing functional richness from 0.17 to 0.31.

Since we assign equal weighting to all pixels (no abundances), we calculated divergence (FDiv) based on Villéger et al.[41] as follows:

$$\Delta|d| = \sum_{i=1}^{S} \frac{1}{S} \cdot |dG_i - \overline{dG}|, \qquad (5)$$

$$\mathrm{FDiv} = \frac{\overline{dG}}{\Delta|d| + \overline{dG}}, \qquad (6)$$

where $S$ is the number of pixels mapped in the functional space, $dG_i$ is the Euclidean distance between the $i$th pixel and the centre of gravity and $\overline{dG}$ is the mean distance of all pixels to the centre of gravity. In this specific case, a functional divergence of 1 would mean that all pixels lie on a sphere with equal distance to the centre of gravity (Supplementary Fig. 15).

The functional evenness index (FEve) was calculated based on the minimum spanning tree (Fig. 2). A distance matrix with Euclidean distances between all the points in the functional space was the basis for deriving the minimum spanning tree using the algorithm of Prim[75] (graphminspantree, Matlab). Finally, evenness was calculated following Villéger et al.[41]:

$$\mathrm{PEW}_l = \frac{\mathrm{EW}_l}{\sum_{l=1}^{S-1} \mathrm{EW}_l}, \qquad (7)$$

$$\mathrm{FEve} = \frac{\sum_{l=1}^{S-1} \min\left(\mathrm{PEW}_l, \frac{1}{S-1}\right) - \frac{1}{S-1}}{1 - \frac{1}{S-1}}, \qquad (8)$$

where $\mathrm{EW}_l$ is the Euclidean distance of branch $l$ in the minimum spanning tree, PEW is the partial weighted evenness and $S$ is the number of pixels mapped in the functional space. Thus $S-1$ corresponds to the number of branches in the minimum spanning tree. A weighting by species abundance is not necessary when mapping pixels, since abundance is inherent in the data (Supplementary Fig. 15).

**Scaling**. To calculate the functional diversity indices for the whole forest, we used a moving window approach (Supplementary Fig. 16). This means that the index values were calculated for each pixel by iterating through all pixels of the functional trait maps. Since diversity is always measured within a certain geographical unit, we used a radial neighbourhood of pixels to calculate the indices. Therefore, the initial pixel size of $6 \times 6$ m of the functional trait maps corresponds to the grain, whereas the neighbourhood of pixels corresponds to the extent (Supplementary Fig. 16). We calculated the diversity indices for an increasing neighbourhood of 6–1,020 m radius with steps of 6 m, resulting in an extent ranging from 113 to $3.27 \times 10^6$ m$^2$. To derive diversity–area curves, we averaged the index values of all forested pixels for each of the 170 extents. For display in Figs. 5–7 and visual assessment, we applied a circular averaging filter (fspecial, disk, Matlab).

**Null models**. We created a null model of randomly distributed trees, or here pixels, to test if the functional traits distribution follows a random distribution, over- or under-dispersion. For each tree or pixel, we kept the traits relationship among the three morphological and physiological traits constant. We then reshuffled the pixels to create random distribution in geographic space (rand, Matlab). Opposed to randomly distribute each trait individually, the trait relationships still hold in the null model. However, there is no spatial autocorrelation any more.

A second null model is used to simulate maximal under-dispersion, which could be resulting from maximal environmental filtering. In this case, we assume that neighbouring pixels in geographic space are also neighbours in functional trait space. For each pixel, it is not the neighbouring pixels in a certain radius which are used to calculate the diversity indices. Instead, the same number of neighbouring pixels are selected from the trait space according to minimal Euclidean distance. This results in a purely theoretical null model, where closest neighbours in geographic space would be closest neighbours in trait space.

**Statistical analysis**. We tested whether patterns of functional traits and trait diversity can be explained by abiotic factors related to topography, soil and radiation (see Supplementary Fig. 4 and Supplementary Table 1). To account for spatial autocorrelation, we used a spatially simultaneous autoregressive error model estimation based on first order neighbours (R package spdep, errorsarlm[76]) to fit a generalized linear model.

Subsequent analysis of variance (ANOVA) with type-I sum of squares was performed at 60 m radius scale. The forest was sampled using 467 pixels projected on a regular grid such that their circular neighbourhood areas did not overlap and remained fully within forest boundaries. Continuous explanatory variables were averaged within 60 m radius, whereas simple majority was used for categorical variables. Continuous explanatory variables were altitude, slope, soil depth and amount of rocky materials. The categorical variable aspect was subdivided in three categories, namely north, south and flat slopes. Curvature was grouped in categories valley, ridge and flat areas. Soil type consisted of eight soil classes (*Dystric Cambisols*, *Luvisols*, *Endogleyic Cambisols*, *Stagnic Cambisols*, *Cambisols*, *Calcic Cambisols*, *Leptosols* and *Regosols*, see Supplementary Fig. 5). Supplementary Figure 4 shows the variance explained based on type-I sum of squares of soil (top panels) and topography (bottom panels), as well as additionally explained factors when adding topography or soil, and radiation to the model. Within groups, the order of the explanatory variables was kept constant. For Supplementary Table 1, the order of the explanatory variables related to topography were determined by the significance when tested individually, with the most significant used first in the combined model.

**Data availability**. The data that support the findings of this study are available from the corresponding author upon reasonable request. An example of the airborne laser scanning and imaging spectroscopy data is available at http://www.geo.uzh.ch/microsite/3dveglab/eod/ for a subset of $300 \times 300$ m. Community and soil data has to be requested directly from the Swiss cantons Zurich or Aargau.

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

## Acknowledgements

This study has been supported by the University of Zurich Research Priority Program on Global Change and Biodiversity (URPP GCB). The research carried out at the Jet Propulsion Lab, California Institute of Technology, was under a contract with the NationalAeronautics and Space Administration. Government sponsorship is acknowledged. Part of the work is based on output from the working group of the National Center for Ecological Analysis and Synthesis on 'Prospects and priorities for satellite monitoring of global terrestrial biodiversity'. Data processing was supported by S3IT (University of Zurich), with assistance of Riccardo Murri and Sergio Maffioletti. We thank Carla Guillén Escribà, Irene Garonna, Walter Jetz, Reinhard Furrer, Meinrad Abegg and Daniel Kükenbrink for fruitful discussions on aspects of the manuscript.

## Author contributions

F.D.S., F.M., B.S., O.L.P., and M.E.S. designed research; F.D.S., B.S., and M.E.S. performed research; F.D.S., F.M., A.H., and M.E.S. analysed data; and F.D.S., F.M., B.S., O.L.P., A.H., D.S.S. and M.E.S. wrote the paper.

## Additional information

**Competing interests:** The authors declare no competing financial interests.

