## [Peer Review File · Nature Communications]

Reviewers' comments:

Reviewer #1 (Remarks to the Author):

Overall this manuscript is a well written and interesting effort at combining imaging spectroscopy and lidar to look at physiological and morphological traits across a site in Switzerland. I agree with the general conclusions and the sentiment of the paper that these emerging remote sensing tools can tell us volumes about biodiversity. The main conclusion, however, as described in the abstract - that diversity was driven by disturbance and harsh environmental conditions, is qualitative at best, given the approaches used in the paper. I describe this concern in more detail below. I think this paper would be much stronger if it included some field validation of the spectral indices, and/or a more compelling suite of indices, and more quantification of the landscape in question.

My main concern with this manuscript is the total absence of field validation of either the spectral indices or the 'conclusions' about specific parts of the landscape and drivers. Regarding the spectral indices, quite a bit of progress has been made in this field since the methods papers referenced in this ms (Gitelson et al 2006 for CHL and CAR and Underwood et al 2003 for LWC), and, importantly, the Gitelson paper's indices were developed based on leaf spectra (not airborne), and the Underwood index for LWC is based on 'image inspection' of an AVIRIS collection in California with very different vegetation than is found in Switzerland. These spectral indices are no more or less robust than the many other hyperspectral indices available, but that doesn't mean much. They also are not particularly compelling as a suite of traits for 'assessing functional diversity' - given the abundance of work on trait axes, focusing on foliar nitrogen and leaf mass per area would make much more sense (e.g. Díaz et al 2016) and are readily mappable with imaging spectroscopy (e.g. Lepine et al 2016, Serbin et al 2014 and Singh et al 2015). Using partial least squares regression (PLSR; as these papers do) to map traits would require field work, but would lead to a more robust statement about actual trait variation. I have very little faith that the three spectral indices used here have substantial correlations to the same measurements from the field at this site, though it's impossible to assess in this paper - one piece of evidence, though, is that there are many zero values in Figs S1 and S2, which suggests either the indices aren't great or the image wasn't properly masked to screen out non-veg pixels (which could seriously impact your conclusions about functional richness - see below). At minimum, if you have plot data with composition information you could use area weighted means of published trait values for the species from something like the TRY database (Kattge et al 2011) to at least know if your indices and therefore trait patterns are in the ballpark of the real values, or if someone else has done a field to index comparison for these indices in or near this site, that would help too. My own experience with hyperspectral vegetation indices applied outside of where they were developed is that they are typically weakly correlated with the actual field measures, but sometimes not correlated at all or even negatively correlated, hence the widespread focus on techniques like PLSR.

Similarly, much of the Discussion is focused on qualitative assessments of the correspondence between the morphological and physiological traits and the landscape being studied. Statements like lines 121-122 saying blue areas of the morphological trait map are due to disturbance and lines 129-131 saying the pink and orange areas of the physiological trait map are beech read as speculation given that there's no information about what's actually in the field. If there were (again) some plot data for the site this could be resolved and these differences could be quantified, but as the paper is written currently we are left to simply trust that the authors know the site well enough to make these blanket statements. They very well may, but I found myself asking a lot of questions - are ALL the blue areas on the morphological map due to the 1999 storm? etc. Even a land cover map, if one exists for the area, could be used to ask these questions in a more quantifiable way, as was done by Dahlin et al 2013.

Below are some additional specific comments and places illustrating the broader points above.

line 7: "informing on" -> "informing"

lines 13-15: "On the one hand" and "on the other hand" or similar are used 4 times in this paper - too many in my opinion.

line 18: what does 'and their gaps' refer to?

line 22: 'environmental filtering, limiting similarity, or neutral theory' - it would be nice to cite the originators of these ideas, not just a review.

lines 44-45: worth mentioning that convex hull volume is a range measure - depends on outermost values - so if you have outliers/nonveg pixels (as you appear to from Figs S1 and S2) that could dramatically alter your CHV and therefore functional richness.

lines 59-60: 'Are forest communities structured by ...?' do they have to be mutually exclusive?

line 62 - Fig 3. What are all the masked areas? I'm guessing agriculture, but this isn't described anywhere. How did you mask non-veg pixels?

lines 62-66: relying on visual interpretation of an RGB image like this is a difficult way of presenting these results. They're described as though you've classified the images but in reality in an RGB it's tough to see these colors clearly.

lines 78-79: why correlate the morpho and physio richness? (and other measures)? what does that tell us? is that related to your overall questions? also describing a correlation plot that isn't shown - could these go in SI?

lines 81-end of methods: Any stats for any of these differences? a lot of these differences are very small (2.7-5.7% for example in line 82) - are they statistically significant? I think an ANOVA or a t-test could be used to check these things.

line 101 - Fig 7: I realize it would make a messy graph, but error bars on the A, B, & C lines in this plot would help clarify whether these results are really outside of the noise.

line 103: 'Black lines' -> 'Solid black lines'

line 111-112: these are the correlations between the actual line and a true log curve?

lines 120-128: a lot of this description should go in the site description, not the discussion.

lines 129-135: any references for these site characteristics?

lines 131-132: 'on one hand'/'on the other hand'

line 143: 'On the other hand'

lines 156-170: These paragraphs seems like they should be in the methods or intro.

lines 187-188: 'On one side'/'On the other side'

line 212: I'm curious how much the 'null model' line would drop if you removed all the pixels with zero values for the six traits from this analysis (as shown in Fig S1). If you trust your metrics, anything with zeros for any of these metrics/indices shouldn't be a plant, so it shouldn't be going in to your diversity metrics. And the abundance of these zeroes means they could really be inflating your null diversity models.

lines 218-219: 'Our results show that...' - this sentence isn't particularly illuminating.

lines 225-226: 'were able to validate these measurements against in-situ community data' - where?

lines 228-229: 'High functional diversity...' again - this reads as speculation given the lack of in-situ data presented.

lines 224-231: While I agree with the sentiments of the conclusion, overall it is lacking in strong conclusions from this actual paper, just speculation about what could be done with more data.

Dahlin, K. M. et al. 2013. Environmental and community controls on plant canopy chemistry in a Mediterranean-type ecosystem. - Proc. Natl. Acad. Sci. U. S. A. 110: 6895-900.

Díaz, S. et al. 2016. The global spectrum of plant form and function. - Nature 529: 167-171.

Kattge, J. et al. 2011. TRY - a global database of plant traits. - Glob. Chang. Biol. 17: 2905–2935.

Lepine, L. C. et al. 2016. Examining spectral reflectance features related to foliar nitrogen in forests: Implications for broad-scale nitrogen mapping. - Remote Sens. Environ. 173: 174–186.

Serbin, S. P. et al. 2014. Spectroscopic determination of leaf morphological and biochemical traits for northern temperate and boreal tree species. - Ecol. Appl. 24: 1651–1669.

Singh, A. et al. 2015. Imaging spectroscopy algorithms for mapping canopy foliar chemical and morphological traits and their uncertainties. - Ecol. Appl. 25: 2180–2197.

Reviewer #2 (Remarks to the Author):

This manuscript is overall clear and well written; the focus on functional diversity is up to date and challenging.

The paper, as it stands, is still difficult to be followed in detail and the following major points should be considered:

- ACRONYMS: The use of acronyms to indicate functional variables renders the paper difficult to read in some parts.

- BACKGROUND THEORY: Major explanations are also needed in some of the figures to be fully understood. As an example, Figure 1 shows graphs which are not rooted in the text and the data being used are not described in detail. In this view, this figure is very difficult to read.

- DIVERGENCE VS. EVENNESS: Functional divergence versus functional evenness are not described in detail apart from some sentences. This renders the whole paper a bit technical and it might better focus on such concepts overall from a theoretical point of view.

- CASE STUDY PRESENTED: The case study might be cool. However, as a reader, I feel that the part describing it is too long with several figures which hamper to catch the real take home message. moreover, again, the theoretical background must be also strengthened.

- TAXONOMIC DIVERSITY - IMPORTANT. A number of papers have dealt with the relationship between functional and taxonomic diversity. Moreover, a number of papers attempted to estimate taxonomic diversity from remote sensing considering both alpha and beta diversity (and also richness versus evenness). This paper does not take into account such approaches. As an example, a good conclusion of the ms might be based on an explicit statement about the power of functional versus mere taxonomic diversity and what is its added values, together with a critique on the use of RS in this framework. At the time being the final take home message is still a bit unclear and technical.

- THEORY BEYOND THE PRESENTED CURVES: No asymptote is reached in Figure 7. Does this mean that total diversity has not been finally reached despite the radius being considered, or, theoretically, in this case the asymptote has no ecological meaning like in rarefaction curves (see Gotelli and Colwell)?

- FLOW: I feel that this paper might benefit from a flowchart addressing the whole analysis, variables, concepts.

- A CRITIQUE TO SHANNON WEAVER THEORY APPLIED TO BIOLOGICAL DIVERSITY: one of the main problems of using Shannon-Weaver theory is that fhd might be high from RS data despite the real values of the RS images. As an example imagine a vector of values [1,2,3,4,5,6,7,8,9] and another one like [1,150,70,25,250,255,44,200,100]. The two vectors will attain exactly the same fhd. This is an important issue. Moreover, fhd might represent both richness and relative abundance with no chance to distinguish them from the final metric. This might be a problem too, but it is not addressed explicitly in the ms.

Reviewer #3 (Remarks to the Author):

The manuscript from Schneider et al uses metrics of functional trait diversity, based on six remotely sensed functional traits, detected at very high spatial resolution, to reveal ecological insights in a Swiss forest system. Using hyperspectral remote sensing at high spatial resolution and at multiple scales to understand plant function and plant functional diversity is a critical advance in ecology. This is the first study to develop an approach to measure remotely sensed plant functional diversity and apply it empirically. The methods developed here are very compelling and relevant to continuous monitoring of ecological changes with global climate change. Both the quality of the data and the analyses are quite high. In short, both the novelty of the work and the quality of it merit publication in a high impact journal like Nature Communications.

However, the paper falls short in developing a believable framework for linking functional trait diversity to community assembly processes. The introduction is very light on explaining how these metrics can be related to community assembly processes, citing good papers, but not actually using the methods proposed in those papers. Much has been written in the community ecology literature about the difficulties of discerning processes from pattern, and these issues require attention. Some convincing efforts have been made in the literature to build a logical hypothesis testing framework for making the leap from pattern to process, and a logical hypothesis testing framework would help this paper. E.g., "If we find pattern x, we can infer process y (and not z) because..."

The best that I can make out is an attempt to do this using the null models, which can sometimes be designed to tease apart, or rule out, alternative interpretations of pattern. Here they are used to identify under- or overdispersion patterns in functional diversity. Those patterns are then interpreted in an oversimplified manner, equating functional similarity with environmental filtering and functional overdispersion with competition. There are many, many processes that can give rise to the same patterns, and so I don't feel the paper is as thoughtful as it needs to be in making inferences about community assembly or other ecological processes. A classical functional trait paper would not be able to draw sweeping conclusions about the respective roles of environmental filtering and limiting similarity based solely on patterns of over and underdispersion functional diversity. Here we have a novel measurement approach, but the same issues are relevant. I believe the authors have a framework in mind that needs to be better articulated. For example, what are the inferences that are believed to be possible when diversity patterns and richness-area relationships are similar for morphological and physiological traits? Lay this out in the introduction so it is clear why the comparison is important.

I will give a specific example from the manuscript about concerning the difficulty of inferring community assembly processes from pattern. Line 212 states environmental filtering is the predominant assembly process, since it is consistently below the null model of randomly distributed traits at all radii. However, line 177 clarifies that FRic for the physiological traits is mainly driven by the difference between conifers and broadleaved trees. And in line 136, we learn that conifers were largely planted by humans. So the low FRic relative to a null model may indicate the presence of only

conifers or only angiosperms, which may be an anthropogenic phenomenon, influenced by decisions humans made about where they thought conifers would best be planted.

The term environmental filtering itself is very vague and much more meaningful when we have a sense of what the environmental factor is that is driving the vegetation pattern. Can the authors marshal more evidence as to which factors in the environment are causing filtering of the vegetation and how the filtering process might operate?

The null models themselves are difficult to follow because their description is so brief. The supplement did not offer further explanation. These should be explained carefully so that the reader can decipher how the null models are being used to discern non-random patterns that reveal something about process.

On line 166-168, the authors state that FRic can indicate that trees are assembled following the principle of limiting similarity, which leads to overdispersion due to direct competition between trees. Again, there are many different processes beyond competition that can lead to overdispersion, and some important studies have shown that competition can actually lead to clumping patterns and underdispersion. Empirically, there is not a clear link. In this study, certainly dispersal processes and phenotypic variation that accompanies ontogenetic changes can contribute to the overdispersion found in the gaps. The authors in fact conclude in line 228, that high functional diversity was related to the occurrence of disturbance areas and patches with mixtures of evergreen coniferous and deciduous broadleaf trees. So in the end, they do not equate overdispersion with competition. At times, the authors apply frameworks from other studies to make ecological inferences, and at other times, they disregard these frameworks and make inferences based on their understanding of a well-studied system. So there seems to be awareness of the issue.

I suggest the authors rewrite to 1) set up a hypothesis testing framework so we can see how inferences are being made, 2) be very careful about the problem of making inferences about process from pattern, and 3) tone down the definitive conclusions drawn about environmental filtering and explain what information would be required to show this. For example, more convincing arguments for environmental filtering would include the relationship between trait values and the environment – surely the data is all there already to do this; linking physiology to performance (as in [19]), if performance measures can be derived from some of the morphological traits at multiple time intervals; looking at changes in physiology and function over time with changes in environment.

The study could also link functional diversity to ecosystem processes. This would be quite exciting.

The scale dependence of diversity is quite interesting in this paper and more could be done with that.

A few smaller points:

Line 108: "A simulated distribution of traits following the assumption of underdispersion where trees being close in functional space are assumed to be close in geographic space, leads to a very low functional richness at all scales." Please clarify exactly how this was done, somewhere.

Line 301-317: In contrast, I am not sure the equations for all the functional diversity metrics need to be included in the methods (can go in the supplement), since they are the same as in the original publication [34]. Note some authors argue there are better metrics of functional diversity (see Scheiner et al). However, I think these are fine.

Line 306: Without defining the particular niche concept applied, calling FRic a measure of niche extent is not clear.

Line 241: Trees of 165 years or greater are not necessarily "old-growth trees." They are mature trees that are old. Unless this is a forest not used by humans (or barely used) for millennia, they would not, strictly, be called old growth.

Line 263: Give definition of plant area index somewhere.

Lines 229-232: "Extending the scale of investigation to individual trees and globally will help to improve our understanding of the interactions of species and traits including genetic, phylogenetic and functional diversity, ultimately allowing to monitor functional diversity from space."

The idea of extending the scale globally is excellent. Throwing in the genetic and phylogenetic diversity is gratuitous; it is not clear what is meant or how it would be done based what is in the manuscript. It seems more important to focus on how ecological inferences would be made globally, based on functional diversity patterns. Surely, emphasizing a temporal component that would allow observation of changes over time, and pairing functional data with environmental data, are critical. It would be nice if this paper could make that case.

Reviewers' comments:

Reviewer #1 (Remarks to the Author):

Overall this manuscript is a well written and interesting effort at combining imaging spectroscopy and lidar to look at physiological and morphological traits across a site in Switzerland. I agree with the general conclusions and the sentiment of the paper that these emerging remote sensing tools can tell us volumes about biodiversity. The main conclusion, however, as described in the abstract - that diversity was driven by disturbance and harsh environmental conditions, is qualitative at best, given the approaches used in the paper. I describe this concern in more detail below. I think this paper would be much stronger if it included some field validation of the spectral indices, and/or a more compelling suite of indices, and more quantification of the landscape in question.

We appreciate the reviewer's interest in our approach to map functional diversity from remotely sensed forest traits. We are thankful for the recommendations to include both, a more quantitative approach as well as additional field-based information and data on environmental conditions of the study site. We have included those suggestions and refer to additional results in the main text and back them up using additional supplementary figures and tables, as described in detail in the remarks below.

My main concern with this manuscript is the total absence of field validation of either the spectral indices or the 'conclusions' about specific parts of the landscape and drivers. Regarding the spectral indices, quite a bit of progress has been made in this field since the methods papers referenced in this ms (Gitelson et al 2006 for CHL and CAR and Underwood et al 2003 for LWC), and, importantly, the Gitelson paper's indices were developed based on leaf spectra (not airborne), and the Underwood index for LWC is based on 'image inspection' of an AVIRIS collection in California with very different vegetation than is found in Switzerland. These spectral indices are no more or less robust than the many other hyperspectral indices available, but that doesn't mean much.

We appreciate this comment and clarify our approach by including three-fold validation approaches. We agree with the general fact, that indices must be treated with care. Indices, applied independently, may risk high collinearity. PLSR may risk infeasible physical solutions, and MCMC approaches may risk too high costs of informative priors needed. Progress has been made on all three aspects. We here use indices with demonstrated applicability in our temperate forest example. We validate leaf-level spectral signatures of 50 beech trees located within the study site (per tree, nine sunlit leaves of three branches at the top of the tree canopy were sampled) using a field measured trait (SPAD meter for Chl). Jointly with data extracted from the ANGERS database¹ (lab measured traits and leaf optical properties), we demonstrate the applicability of spectral indices at leaf level. We also simulated canopy spectra using the 3D radiative transfer model DART (Schneider, et al. 2014) for the exact illumination/observation angles during airborne data acquisitions, by using the above leaf optical properties (in-situ measurements and ANGERS database). We then derived chlorophyll, carotenoids and equivalent water thickness for the simulated canopy spectra using the same approach as for the remotely sensed canopy spectra at various illumination angles. Finally, we compared all validation approaches and conclude that frequency distribution, magnitude, and trait correlations of measured and simulated spectra correspond well ($r^2 > 0.5$) with our airborne data. We added a paragraph in the manuscript on lines

¹ <http://opticleaf.ipgp.fr/index.php?page=database>

115-124 as well as in the supplementary information on lines 10-31 and additional figures (Supp. Figs 7-8).

They also are not particularly compelling as a suite of traits for 'assessing functional diversity' - given the abundance of work on trait axes, focusing on foliar nitrogen and leaf mass per area would make much more sense (e.g. Díaz et al 2016) and are readily mappable with imaging spectroscopy (e.g. Lepine et al 2016, Serbin et al 2014 and Singh et al 2015).

We fully concur with this comment – we do not claim general applicability of those 6 traits beyond the study site. Currently, there is little convergence (in the remote sensing community) on the optimal choice of traits derived from remote sensing and their ability to map functional diversity. We focus on traits that are directly observable using imaging spectrometer reflectance data. The important works of Lepine, Serbin, Singh (and also Asner) use either proxies of proxies (for LMA and N) or in-situ data based calibration methods. We explain our selection of traits and their ecological relevance in more detail in the introduction (lines 55-63). Though very important, the Díaz et al. 2016 study is focusing on trait variation worldwide. Our method will be extended to using more traits in the future. Currently, our 6 traits feed three diversity measures, resulting in an over-determined system. The inverse (how well these 6 traits actually describe functional diversity at larger scales (and across biomes)) has not been assessed here and is subject to future work.

Using partial least squares regression (PLSR; as these papers do) to map traits would require field work, but would lead to a more robust statement about actual trait variation. I have very little faith that the three spectral indices used here have substantial correlations to the same measurements from the field at this site, though it's impossible to assess in this paper - one piece of evidence, though, is that there are many zero values in Figs S1 and S2, which suggests either the indices aren't great or the image wasn't properly masked to screen out non-veg pixels (which could seriously impact your conclusions about functional richness - see below).

Scaling traits between leaf and canopy level is a main challenge (especially for traits related to leaf water or nitrogen). We refer to the discussions by Knyazikhin, et al. 2012 and subsequent replies, stating the limitations of PLSR. In our approach, we combined data acquisitions at different illumination/observation angles and used spatial aggregation leading to a reduction of shadow and illumination/structure effects, which is crucial to be able to apply spectral indices at canopy level (statement added on lines 118-120). Regarding zero values, we used confusing terminology. In fact, zero values reflect scaling by which minimum values of vegetated pixels were converted to zero. We are thankful for this comment and include a more comprehensive description of how forested pixels were defined (lines 413-417).

At minimum, if you have plot data with composition information you could use area weighted means of published trait values for the species from something like the TRY database (Kattge et al 2011) to at least know if your indices and therefore trait patterns are in the ballpark of the real values, or if someone else has done a field to index comparison for these indices in or near this site, that would help too.

Thank you for this suggestion. Indeed, we have a core study site at the Laegern forest of approximately 5.5 ha, where we measured location and taxonomic identity of all trees with a DBH above 20 cm (resulting in 1307 trees of 13 different species). We used these plot data to calculate

community-weighted means using the plot-level species abundances and species-level trait values from the TRY database. Although TRY is not suitable for assessing intra-specific trait variation and is ignoring trait plasticity, we find a positive relationship to index values for chlorophyll and equivalent water thickness (lines 116-118). Unfortunately, there are not enough values in TRY for carotenoids, making it impossible to apply gap-filling and compare to the remotely sensed trait values. In addition, we added Supplementary Figure 5 (on trait ranges) and Figure 6 (remote sensing vs. TRY) based on this suggestion.

My own experience with hyperspectral vegetation indices applied outside of where they were developed is that they are typically weakly correlated with the actual field measures, but sometimes not correlated at all or even negatively correlated, hence the widespread focus on techniques like PLSR.

Additionally to the above-mentioned points, we also include a comparison of trait ranges from the remotely sensed trait maps to trait ranges simulated based on Schneider, et al. 2014 with independently measured leaf optical properties (LOP). The model was used with the same in-situ LOPs, averaged for deciduous broadleaf and coniferous needle trees, respectively, and parameters as described in Schneider, et al. 2014. The traits occur in a similar part of the whole trait range of the forest (see lines 115-116 and Suppl. Fig. 5). The variation in remotely sensed traits is caused by the actual physiological trait variation between individuals within and among species. PLSR methods may be superior to index-based approaches, in particular where models used reflect well the canopy architecture. We used inversion schemes with SLC (Hapke soil model, PROSPECT, 4SAIL2) based on PLSR and Bayesian approaches (Laurent et al. 2013, 2014) and found trait retrievals to be worse in forests than in optimal 'turbid medium' scatterer (such as agricultural canopies). We therefore used the forward model approach based on DART, allowing to independently validate trait retrievals in 'sub-optimal turbid medium' architectures, such as this temperate forest.

Similarly, much of the Discussion is focused on qualitative assessments of the correspondence between the morphological and physiological traits and the landscape being studied. Statements like lines 121-122 saying blue areas of the morphological trait map are due to disturbance and lines 129-131 saying the pink and orange areas of the physiological trait map are beech read as speculation given that there's no information about what's actually in the field. If there were (again) some plot data for the site this could be resolved and these differences could be quantified, but as the paper is written currently we are left to simply trust that the authors know the site well enough to make these blanket statements. They very well may, but I found myself asking a lot of questions - are ALL the blue areas on the morphological map due to the 1999 storm? etc. Even a land cover map, if one exists for the area, could be used to ask these questions in a more quantifiable way, as was done by Dahlin et al 2013.

We agree that plot data or a landcover map as well as additional environmental variables would help to support the qualitative assessment and make the manuscript stronger. Therefore, we add forest stand polygon data from the Cantons of Aargau and Zurich with information on the most dominant species and juvenile forest patches (c.f., Suppl. Fig. 2) and for quantitative statements (lines 95-99, 106-110, 242-245). We reformulate the first paragraph of the discussion (lines 181-186) to be more precise about disturbance areas. For an additional 'quantification of the landscape', we include the topographic variables altitude, slope, aspect and curvature and provide additional ANOVA tests (lines 131-144, 147-149, 153-156, Suppl. Tab. 1). Finally, we add figures on

topography, radiation and soil variables (Suppl. Fig. 10) and demonstrate a significant difference between the ridge and lower altitudes (Fig. 7, lines 231-238, Suppl. Tab. 2). See also comments of reviewer 3.

Below are some additional specific comments and places illustrating the broader points above.

line 7: "informing on" -> "informing"

Corrected on line 27.

lines 13-15: "On the one hand" and "on the other hand" or similar are used 4 times in this paper - too many in my opinion.

Thank you for the comment, we removed or replaced the phrase except for lines 33-36.

line 18: what does 'and their gaps' refer to?

It should refer to the data gaps in in-situ data. We clarified the sentence on line 38.

line 22: 'environmental filtering, limiting similarity, or neutral theory' - it would be nice to cite the originators of these ideas, not just a review.

We removed this sentence (due to another reviewer comment) and added the respective references later in the text on lines 225-228.

lines 44-45: worth mentioning that convex hull volume is a range measure - depends on outermost values - so if you have outliers/nonveg pixels (as you appear to from Figs S1 and S2) that could dramatically alter your CHV and therefore functional richness.

Indeed, that is right. We now mention this on line 68.

lines 59-60: 'Are forest communities structured by ...?' do they have to be mutually exclusive?

We agree that they are not exclusive. We rephrase the question on lines 89-90.

line 62 - Fig 3. What are all the masked areas? I'm guessing agriculture, but this isn't described anywhere. How did you mask non-veg pixels?

We add a paragraph in the Methods on lines 413-417, describing how we derived the forest mask.

lines 62-66: relying on visual interpretation of an RGB image like this is a difficult way of presenting these results. They're described as though you've classified the images but in reality in an RGB it's tough to see these colors clearly.

We prefer to show the trait maps using continuous (RGB) color-scales, since it corresponds to the 3D trait space definition and how the traits are used to calculate the FD indices. However, we agree that our classification for interpretation purposes is not optimal (definition given in methods lines 406-412). We updated the graph by adding Supplementary Figure 2, where the main classes are shown and compared to plot data.

lines 78-79: why correlate the morpho and physio richness? (and other measures)? what does that tell us? is that related to your overall questions? also describing a correlation plot that isn't shown - could these go in SI?

Following other reviewer's comments (see below) we added a statement in the introduction (lines 83-87) and the conclusions (lines 307-311) why the comparison between patterns of morphological and physiological is relevant.

lines 81-end of methods: Any stats for any of these differences? a lot of these differences are very small (2.7-5.7% for example in line 82) - are they statistically significant? I think an ANOVA or a t-test could be used to check these things.

We added results of an ANOVA test in the manuscript (lines 131-144, 147-149, 153-156) and the full ANOVA table in Supplementary Table 1.

line 101 - Fig 7: I realize it would make a messy graph, but error bars on the A, B, & C lines in this plot would help clarify whether these results are really outside of the noise.

The A, B, C lines are from a single pixel in the center of the subregions. We apologize for the misunderstanding and clarify on lines 161-163 and in the caption of Fig. 8. We agree that error bars for the whole subregions would be helpful but a bit messy, why we add it to Supplementary Figure 9.

line 103: 'Black lines' -> 'Solid black lines'

Changed on line 163.

line 111-112: these are the correlations between the actual line and a true log curve?

We clarified that it is with regard to the richness-area relationship with area on the x-axis (lines 171-173). The function used to fit is: $y = a \cdot \log(x) + b$

lines 120-128: a lot of this description should go in the site description, not the discussion.

We provide description only relevant for the discussion of the functional traits and trait diversity maps. We prefer to leave this section unchanged, otherwise the context of our discussion might be lost.

lines 129-135: any references for these site characteristics?

Additional data is described in a new paragraph 'Environmental data' in Materials and Methods (lines 353-364), added to Results as well as Supplementary Figures 2 and 10.

lines 131-132: 'on one hand'/'on the other hand'

Changed on lines 195/197.

line 143: 'On the other hand'

Changed on line 208.

lines 156-170: These paragraphs seems like they should be in the methods or intro.

We move the first paragraph to Methods (lines 419-425) and rephrase the second paragraph (lines 222-230).

lines 187-188: 'On one side'/'On the other side'

Changed on lines 254-255.

line 212: I'm curious how much the 'null model' line would drop if you removed all the pixels with zero values for the six traits from this analysis (as shown in Fig S1). If you trust your metrics, anything

with zeros for any of these metrics/indices shouldn't be a plant, so it shouldn't be going in to your diversity metrics. And the abundance of these zeroes means they could really be inflating your null diversity models.

We agree that an absolute value of 0 in any of the traits would not be a plant (or an infeasible retrieval). However, zero values result from scaling index values from 0 to 1, which do not include absolute 0 values. Therefore, our scaling index values do influence absolute values (maximum richness), but do not show an influence on patterns and shapes of the curves.

lines 218-219: 'Our results show that...' - this sentence isn't particularly illuminating.

We rephrase the paragraph on lines 293-297 by adding an example of the scalability of the method to coarser spatial resolutions.

lines 225-226: 'were able to validate these measurements against in-situ community data' - where?
lines 228-229: 'High functional diversity...' again - this reads as speculation given the lack of in-situ data presented.

lines 224-231: While I agree with the sentiments of the conclusion, overall it is lacking in strong conclusions from this actual paper, just speculation about what could be done with more data.

We hope to have resolved these three issues by adding additional plot data, extending the results and supplementary information as well as rephrasing the conclusions (lines 301-320), see main comments above.

Dahlin, K. M. et al. 2013. Environmental and community controls on plant canopy chemistry in a Mediterranean-type ecosystem. - Proc. Natl. Acad. Sci. U. S. A. 110: 6895–900.

Díaz, S. et al. 2016. The global spectrum of plant form and function. - Nature 529: 167–171.

Kattge, J. et al. 2011. TRY - a global database of plant traits. - Glob. Chang. Biol. 17: 2905–2935.

Lepine, L. C. et al. 2016. Examining spectral reflectance features related to foliar nitrogen in forests: Implications for broad-scale nitrogen mapping. - Remote Sens. Environ. 173: 174–186.

Serbin, S. P. et al. 2014. Spectroscopic determination of leaf morphological and biochemical traits for northern temperate and boreal tree species. - Ecol. Appl. 24: 1651–1669.

Singh, A. et al. 2015. Imaging spectroscopy algorithms for mapping canopy foliar chemical and morphological traits and their uncertainties. - Ecol. Appl. 25: 2180–2197.

Knyazikhin, Y. et al. 2013. Hyperspectral remote sensing of foliar nitrogen content. PNAS 110(3): E185-E192.

Laurent, V.C.E., et al. 2014. Bayesian object-based estimation of LAI and chlorophyll from a simulated Sentinel-2 top-of-atmosphere radiance image. Remote Sens. Environ. 140: 318-329.

Laurent, V.C.E., et al. 2013. A Bayesian object-based approach for estimating vegetation biophysical and biochemical variables from APEX at-sensor radiance data. Remote Sens. Environ. 139: 6-17

Schneider, F.D. et al. 2014. Simulating imaging spectrometer data: 3D forest modeling based on LiDAR and in situ data. Remote Sens. Environ. 152: 235-250.

Reviewer #2 (Remarks to the Author):

This manuscript is overall clear and well written; the focus on functional diversity is up to date and challenging.

The paper, as it stands, is still difficult to be followed in detail and the following major points should be considered:

- ACRONYMS: The use of acronyms to indicate functional variables renders the paper difficult to read in some parts.

We substantially reduced the use of acronyms and hope to have increased the readability of the manuscript overall.

- BACKGROUND THEORY: Major explanations are also needed in some of the figures to be fully understood. As an example, Figure 1 shows graphs which are not rooted in the text and the data being used are not described in detail. In this view, this figure is very difficult to read.

We extended the explanation and embedded Fig. 1 in the manuscript on lines 68-76 and 428-430.

- DIVERGENCE VS. EVENNESS: Functional divergence versus functional evenness are not described in detail apart for some sentences. This renders the whole paper a bit technical and it might better focus on such concepts overall from a theoretical point of view.

We add some more theoretical background on the concept of functional divergence and evenness on lines 68-76. Although important to include, we found functional divergence and evenness to be less relevant to assess scale-dependent functional diversity than functional richness. We clarify this in the Discussion (lines 280-283) and the Conclusions (lines 311-313).

- CASE STUDY PRESENTED: The case study might be cool. However, as a reader, I feel that the part describing it is too long with several figures which hamper to catch the real take home message. moreover, again, the theoretical background must be also strengthened.

Based on this recommendation, we have reshaped the description as well as the figures to improve readability and not dilute the main message. With all changes proposed by the other reviewers and in particular rewriting the introduction and conclusions (lines 301-320), we feel the take home message to be much clearer now.

- TAXONOMIC DIVERSITY - IMPORTANT. A number of papers have dealt with the relationship between functional and taxonomic diversity. Moreover, a number of papers attempted to estimate taxonomic diversity from remote sensing considering both alpha and beta diversity (and also richness versus evenness). This paper does not take into account such approaches. As an example, a good conclusion of the ms might be based on an explicit statement about the power of functional versus mere taxonomic diversity and what is its added values, together with a critique on the use of RS in this framework. At the time being the final take home message is still a bit unclear and technical.

We explain more clearly in the Introduction the interest why we measure functional diversity independently of taxonomic diversity. We also add additional references on the importance of considering variation within species and possible redundancy of different species with regard to functional diversity (lines 44-47). The focus of this paper is clearly on mapping functional diversity. As mentioned above, we have rewritten the conclusions to sharpen the main message of the paper.

- THEORY BEYOND THE PRESENTED CURVES: No asymptote is reached in Figure 7. Does this mean that total diversity has not been finally caught despite the radius being considered, or, theoretically, in this case the asymptote has no ecological meaning like in rarefaction curves (see Gotelli and Colwell)?

Indeed, the total diversity has not been caught by the largest radius applied. The curve is supposed to reach the asymptote as soon as the maximum range of traits in the ecosystem is captured by a certain neighborhood area. If we were to extend the radius further, we must include landscape level diversity effects (agriculture, urban, permanent grasslands, etc.) and may be able to validate or invalidate if the asymptote has ecological meaning. In our case, the limited site extend will not allow us to do so, but it remains an important and interesting question!

- FLOW: I feel that this paper might benefit from a flowchart addressing the whole analysis, variables, concepts.

We have added a flowchart to the supplementary information illustrating the workflow from the remote sensing measurement to the final diversity maps (line 368-369, Suppl. Fig. 12).

- A CRITIQUE TO SHANNON WEAVER THEORY APPLIED TO BIOLOGICAL DIVERSITY: one of the main problems of using Shannon-Weaver theory is that fhd might be high from RS data despite the real values of the RS images. As an example imagine a vector of values [1,2,3,4,5,6,7,8,9] and another one like [1,150,70,25,250,255,44,200,100]. The two vectors will attain exactly the same fhd. This is an important issue. Moreover, fhd might represents both richness and relative abundance with no chance to distinguish them from the final metric. This might be a problem too, but it is not addressed explicitly in the ms.

Maybe there is a misunderstanding on how we applied the index. We do not calculate relative abundance how it is done for species (Shannon-Index), but we use the proportion of foliage in a respective canopy layer instead (see lines 378-380). So for example the vector p [18%, 19%, 20%, 21%, 22%] with 18% of foliage in the first layer, 19% in the second, and so on, does not have the same FHD value as the vector p [5%, 15% 20%, 25%, 35%]. We agree on the second part of the comment, saying that FHD is not just a measure of how different the layers are but also how many layers there are. We add a comment on this and possible correlation with canopy height in the manuscript on lines 380-382.

Reviewer #3 (Remarks to the Author):

The manuscript from Schneider et al uses metrics of functional trait diversity, based on six remotely sensed functional traits, detected at very high spatial resolution, to reveal ecological insights in a Swiss forest system. Using hyperspectral remote sensing at high spatial resolution and at multiple scales to understand plant function and plant functional diversity is a critical advance in ecology. This is the first study to develop an approach to measure remotely sensed plant functional diversity and apply it empirically. The methods developed here are very compelling and relevant to continuous monitoring of ecological changes with global climate change. Both the quality of the data and the analyses are quite high. In short, both the novelty of the work and the quality of it merit publication in a high impact journal like Nature Communications.

We are happy to see that the reviewer appreciates the novelty and significance of our work, combining remote sensing and ecological methodologies to map and understand patterns of functional diversity. We appreciate the detailed comments and recommendations, and answer point by point as follows.

However, the paper falls short in developing a believable framework for linking functional trait diversity to community assembly processes. The introduction is very light on explaining how these metrics can be related to community assembly processes, citing good papers, but not actually using the methods proposed in those papers. Much has been written in the community ecology literature about the difficulties of discerning processes from pattern, and these issues require attention. Some convincing efforts have been made in the literature to build a logical hypothesis testing framework for making the leap from pattern to process, and a logical hypothesis testing framework would help this paper. E.g., "If we find pattern x, we can infer process y (and not z) because..."

We agree that our interpretation of the described patterns in terms of potential processes underpinning them is too generalizing and partly even speculative. It was actually not intended to be the main message of the paper, neither did we mean that this interpretation should serve as the hypothesis framework. Therefore, we realized that we need to step back from the strong interpretation about processes responsible for patterns and focus more on revealing the patterns (e.g. with regards to trait convergence and divergence). We changed the wording throughout the manuscript and clarified the focus, being more precise about possible reasons causing the observed patterns and relations to environmental factors.

The best that I can make out is an attempt to do this using the null models, which can sometimes be designed to tease apart, or rule out, alternative interpretations of pattern. Here they are used to identify under- or overdispersion patterns in functional diversity. Those patterns are then interpreted in an oversimplified manner, equating functional similarity with environmental filtering and functional overdispersion with competition. There are many, many processes that can give rise to the same patterns, and so I don't feel the paper is as thoughtful as it needs to be in making inferences about community assembly or other ecological processes. A classical functional trait paper would not be able to draw sweeping conclusions about the respective roles of environmental filtering and limiting similarity based solely on patterns of over and underdispersion functional diversity. Here we have a novel measurement approach, but the same issues are relevant. I believe the authors have a framework in mind that needs to be better articulated. For example, what are the inferences that are believed to be possible when diversity patterns and richness-area relationships are similar for

morphological and physiological traits? Lay this out in the introduction so it is clear why the comparison is important.

Indeed, the hypothesis to be tested in the context of functional diversity patterns was the one of over- and underdispersion or trait divergence and convergence, respectively. We adapted the research question on lines 89-90 and the discussion on lines 225-230. As mentioned above in comments to reviewer 1, we now better justify our selection of functional traits and our interest in comparing the two groups of traits. The study of Díaz et al. 2016, among many others, has shown strong correlations among traits. By demonstrating their similar patterns we therefore provide justification that these are mapped correctly and are representative for functional traits in general. We add this to the Introduction (lines 55-63, 83-87) and the Conclusions (lines 307-316).

I will give a specific example from the manuscript about concerning the difficulty of inferring community assembly processes from pattern. Line 212 states environmental filtering is the predominant assembly process, since it is consistently below the null model of randomly distributed traits at all radii. However, line 177 clarifies that FRic for the physiological traits is mainly driven by the difference between conifers and broadleaved trees. And in line 136, we learn that conifers were largely planted by humans. So the low FRic relative to a null model may indicate the presence of only conifers or only angiosperms, which may be an anthropogenic phenomenon, influenced by decisions humans made about where they thought conifers would best be planted.

Thank you for the clarification. We agree that we are observing trait convergence, but cannot make the direct link to environmental filtering especially due to anthropogenic influence in parts of the forest. We changed the manuscript text accordingly and add a statement specifically on lines 225-230, 253-256.

The term environmental filtering itself is very vague and much more meaningful when we have a sense of what the environmental factor is that is driving the vegetation pattern. Can the authors marshal more evidence as to which factors in the environment are causing filtering of the vegetation and how the filtering process might operate?

We realize that a more detailed description of the environment at the studied forest is needed (see also comments above to reviewer 1). By including topographic variables (altitude, slope, aspect, curvature) as well as data on radiation and soil (type, depth, coarse material), we are able to better describe the environmental conditions possibly leading to environmental filtering. Especially at the mountain ridge we can observe trait convergence and reduced diversity, which coincides with significantly different conditions regarding soil (shallower, rockier), radiation (higher) and topography (steeper) than in all other areas. We add this to the discussion on lines 231-238, 242-245 and add Fig. 7, Suppl. Fig. 10, and Suppl. Tab. 1 & 2.

The null models themselves are difficult to follow because their description is so brief. The supplement did not offer further explanation. These should be explained carefully so that the reader can decipher how the null models are being used to discern non-random patterns that reveal something about process.

Since we are stepping back from strongly inferring processes from patterns, we do not include any new null models. The current null models are mainly used to show over- or underdispersion or a

random distribution of traits. We apologize for not being clear enough and therefore extend the description of the null models on lines 466-476.

On line 166-168, the authors state that FRic can indicate that trees are assembled following the principle of limiting similarity, which leads to overdispersion due to direct competition between trees. Again, there are many different processes beyond competition that can lead to overdispersion, and some important studies have shown that competition can actually lead to clumping patterns and underdispersion. Empirically, there is not a clear link. In this study, certainly dispersal processes and phenotypic variation that accompanies ontogenetic changes can contribute to the overdispersion found in the gaps. The authors in fact conclude in line 228, that high functional diversity was related to the occurrence of disturbance areas and patches with mixtures of evergreen coniferous and deciduous broadleaf trees. So in the end, they do not equate overdispersion with competition. At times, the authors apply frameworks from other studies to make ecological inferences, and at other times, they disregard these frameworks and make inferences based on their understanding of a well-studied system. So there seems to be awareness of the issue.

We agree that these interpretations were not made clearly enough. As mentioned in previous comments, we have adapted the argumentation throughout the manuscript.

I suggest the authors rewrite to 1) set up a hypothesis testing framework so we can see how inferences are being made, 2) be very careful about the problem of making inferences about process from pattern, and 3) tone down the definitive conclusions drawn about environmental filtering and explain what information would be required to show this. For example, more convincing arguments for environmental filtering would include the relationship between trait values and the environment – surely the data is all there already to do this; linking physiology to performance (as in [19]), if performance measures can be derived from some of the morphological traits at multiple time intervals; looking at changes in physiology and function over time with changes in environment. The study could also link functional diversity to ecosystem processes. This would be quite exciting.

As mentioned in previous comments, we are stepping back from making strong inferences from patterns about processes. We have added environmental variables and regressions of functional diversity on these variables but prefer not to extend this to regressions of individual traits because we think this is outside the scope of this paper. The focus should lie on revealing and interpreting the functional diversity patterns. We do hope to link functional diversity and ecosystem functioning in future work, by combining remotely sensed traits and forest composition with remotely-sensed ecosystem-function variables. However, this will require a whole new study and manuscript and we feel that including more aspects like productivity measures would make the present paper losing focus.

The scale dependence of diversity is quite interesting in this paper and more could be done with that.

We are very interested in working on the scale dependence of diversity, but we believe that it would exceed the scope and make the manuscript difficult to follow. Future work on this aspect is planned for new submissions. We now do, however, mention the potential to upscale our approach for monitoring diversity from space, in a paragraph at the end of the Discussion (lines 293-297) and a figure in supplementary information (Supp. Fig. 11).

A few smaller points:

Line 108: "A simulated distribution of traits following the assumption of underdispersion where trees being close in functional space are assumed to be close in geographic space, leads to a very low functional richness at all scales." Please clarify exactly how this was done, somewhere.

We extended the description of the null models in the Methods section on lines 466-476.

Line 301-317: In contrast, I am not sure the equations for all the functional diversity metrics need to be included in the methods (can go in the supplement), since they are the same as in the original publication [34]. Note some authors argue there are better metrics of functional diversity (see Scheiner et al). However, I think these are fine.

We included the diversity metrics (formulas) in the main manuscript, because they are crucial to understand the resulting index values and they are not exactly the same due to the lack of abundance weighting (although the formula behind is the same). Thank you for this comment. We are aware of the many diversity metrics, each with advantages and disadvantages. However, since there are already many studies on this topic, we did not want this study to be focused on the selection of index.

Line 306: Without defining the particular niche concept applied, calling FRic a measure of niche extent is not clear.

We extend the description to be clear what we mean by niche (lines 433-434).

Line 241: Trees of 165 years or greater are not necessarily "old-growth trees." They are mature trees that are old. Unless this is a forest not used by humans (or barely used) for millennia, they would not, strictly, be called old growth.

Changed on line 330.

Line 263: Give definition of plant area index somewhere.

The definition is given on lines 374-375.

Lines 229-232: "Extending the scale of investigation to individual trees and globally will help to improve our understanding of the interactions of species and traits including genetic, phylogenetic and functional diversity, ultimately allowing to monitor functional diversity from space." The idea of extending the scale globally is excellent. Throwing in the genetic and phylogenetic diversity is gratuitous; it is not clear what is meant or how it would be done based what is in the manuscript. It seems more important to focus on how ecological inferences would be made globally, based on functional diversity patterns. Surely, emphasizing a temporal component that would allow observation of changes over time, and pairing functional data with environmental data, are critical. It would be nice if this paper could make that case.

Thank you for this suggestion. We agree that it is of high importance to head towards a global system for assessing functional diversity as well as including the temporal component. We try to make that case at the end of the Conclusions on lines 317-320.

Reviewers' comments:

Reviewer #1 (Remarks to the Author):

Overall I think that this manuscript shows much improvement over the previous version and I appreciate the additional work that the authors have put in to it. However, it seems as though much of the additional work has been added to the supplement and (some) mentioned in the results section but has not been described in the methods (e.g. comparisons between field & TRY traits and RS, ANOVA approach) making it difficult to evaluate these additions - there are some descriptions of these new methods in the rebuttal, but they should be in the actual manuscript. A few examples of now unanswered questions: how were the categorical variables handled in the ANOVA? what do the yellow lines in Sup Fig 8 c and d represent? What's the rationale for supplementary table 2? I think this manuscript is moving in the right direction, but much of the continuity has been lost as more analyses have been done but the text has not been appropriately or comprehensively updated.

Reviewer #2 (Remarks to the Author):

I am feeling that the authors did a great job to reply to my comments together with those of the other reviewers. I am also feeling that some of the comments from Reviewer #1 should still be considered in full detail.

E.g. the need of field data to calibrate hyperspectral traits and indices. A more convincing statement should be put in the main text.

Another minor point concerns the Supplementary material. I am feeling that the frequency distribution of FHD is a key concept in this manuscript which might deserve space in the main ms instead of being basically hidden in the Supplementary material. This is also true for trait correlations.

However, this is not only an authors' decision but overall an editorial one.

This said, the paper is robust enough to deserve publication.

Duccio Rocchini

Reviewer #3 (Remarks to the Author):

There remain a few conceptual issues, which require further development.

What is meant by functional diversity and what do we learn from it? Here it seems to be the distribution of several traits that are retrievable from RS indices and can then be mapped. But different traits will give different diversity values, and there should be some context and foundation for the choice - linked to larger conceptual questions, presumably about ecosystem function. The functional diversity angle is weak without a clear conceptual framework or link to ecosystem function.

It seems the integration of diversity indices with lidar is the most novel angle rather than "advanced trait retrieval" and mapping traits from airborne data. The relationship between trait maps and forest types is described, but many forest types can be mapped from imagery without trait mapping. Clarification of how the insights in the ms are novel is needed. How is it novel to apply indices of functional diversity to airborne imagery, and what is learned by doing so?

The ms reveals low functional diversity of mountain ridges and high functional diversity of disturbed areas. What does this tell us about ecosystem function? That disturbed and early successional areas are more productive and high altitudes are less productive?

The relationship between lower functional diversity and higher altitude (ridges) with shallower soils that have fewer resources (nutrients, water) is interesting and could be the dominant focus of the ms. The forest traits may correlate with a detailed soil map, if it exists for the area.

In terms of the way functional diversity is calculated, functional richness, evenness and dispersion are all important components of functional diversity. In addition to the number of functionally distinct units (richness), how different they are (dispersion) and in how much they fill trait space (evenness) matters.

Some concepts in the discussion, including "competitive exclusion", "underdispersion", "divergence/convergence" are mentioned but not well explained. The interpretation remains speculative.

The questions of the manuscript need to have context and to be grounded in a foundation for interpretation. What are the expectation for why trait diversity or dispersion patterns should change with scale and what do we learn when they do or do not shift according to expectations? As posed, three questions set the study up to be fairly descriptive.

- 1) What are the spatial patterns of morphological and physiological trait diversity derived from remotely sensed laser scanning and spectrometer data?
- 2) How does functional diversity change with scale?
- 3) Are most of the forest communities structured by trait convergence, trait divergence or randomly distributed traits?

We need to understand why shifts with scale are important and what the expectations are for the functional diversity to vary with scale and how this relates to the ability of remote sensing to detect functional diversity shifts with scale.

Line 45- 46: Some imprecision in language: "different species can be redundant with regard to their functional diversity" - does this really mean species can be redundant in terms of their functional traits?

Lines 50-55: "Canopy height, density, layering ... influence light availability, resource consumption and species diversity species diversity": Some clarification required here - in the understory? of forest birds? small mammals? what is being referred to?

Line 64: "Quantifying functional diversity from morphological and physiological traits reveals the distribution of species or individuals in the functional trait space". This is not actually true. The authors intend meaning here that is not being conveyed.

Line 66: "Functional richness is calculated as the convex hull volume of the community niche": This does not account for the filling of the convex hull. Richness would be the number of functional "units", and the number of units is independent of the niche volume. Many different units can occupy a small space, but three extreme units can delineate a large space. This is why dispersion and evenness are important in FD. Then one could ask whether all three factors (richness, evenness, dispersion) are needed to explain a certain ecosystem function. This would be an interesting avenue for the ms.

Lines 84-86: Why do similar pattern in traits indicate that they are mapped correctly? Is this because all of the traits are correlated with each other?

"and are representative for functional traits in general" I am not sure what this is supposed to mean. Certainly not all functional traits are correlated with each other.

Lines 86-86 "Nevertheless, we also expect to see differences due to different responses to abiotic factors and expressions of plant health and development" Here is where a conceptual framework laying out expectations is important. What kinds of patterns and deviations are expected and how should they change with scale associated with the processes and functions that predominate at different scales? This would provide a means to interpret otherwise descriptive patterns.

Lines 126-128: "Patterns of morphological and physiological richness exhibit strongest correlation at medium scale between 60 and 240 m radius." Why is the correlation important to test and what would the expectation be?

Lines 280-283: "Functional divergence and evenness are generally high, mainly scale invariant and vary only in a small range, scale-dependency of functional diversity in this forest ecosystem is best represented by functional richness." Without using a functional diversity metric that incorporates all three components and allows examination of the influence of each component separately, this conclusion may not be supported. Or one could investigate the influence of each component of functional diversity and its association with a specific abiotic factor or ecosystem function.

To address the central remaining points raised by Reviewers 1 and 2, we request that you revise the paper to better integrate the field data validations (Supplementary Figures 7-8) with the main text. If you do not think that putting the results/figures themselves in the main text would be possible, please at least add detail on the justification, methods and analyses to the main text. In general, we encourage all methods to be in the main text, to aid the reader.

We appreciate the recommendations and are glad that our additional work has been recognized as an essential improvement of the manuscript. To better integrate the field data validation, we applied three major changes. First, we added a paragraph to the introduction defining the testing of the consistency of our method and the field data validations as one of the main goals of the study. Second, we added more detailed results and discussed them in the sections Results as well as Discussion, but we decided to keep the figures in the supplementary material. Third, we added a detailed description of the field data and methods to the main text. We include a PDF-Version of the manuscript, where related text passages are highlighted in orange. Similarly, we highlighted text in green, yellow and blue, respectively, for the three main points among reviewer 3's comments.

Reviewer #1 (Remarks to the Author):

Overall I think that this manuscript shows much improvement over the previous version and I appreciate the additional work that the authors have put in to it. However, it seems as though much of the additional work has been added to the supplement and (some) mentioned in the results section but has not been described in the methods (e.g. comparisons between field & TRY traits and RS, ANOVA approach) making it difficult to evaluate these additions - there are some descriptions of these new methods in the rebuttal, but they should be in the actual manuscript.

We added the subsections 'Field data' and 'Field validation of physiological traits' to the main manuscript on line 391 and 440, describing additional data and methods in more detail.

A few examples of now unanswered questions:

How were the categorical variables handled in the ANOVA?

We added an additional section on 'Statistical analysis' in the Methods section on line 541, explaining the ANOVA approach and a detailed description of the independent categorical variables. With regard to Supplementary Tab. 2, we realized that ANOVA is not the suitable statistical analysis when both dependent and independent variables are categorical. We replaced this table by Supplementary Fig. 9, showing the results of a variance partitioning based on soil, topography and radiation.

What do the yellow lines in Sup Fig 8 c and d represent?

We added clarification to the caption of the Figure. The proposed spectral index to derive carotenoids seems not to be suitable for very high carotenoids values. Since we do not find very high carotenoids values at our study site, we applied a second linear regression on carotenoids values below 15 $\mu\text{g}/\text{cm}^2$.

What's the rationale for supplementary table 2?

We replaced Supplementary Tab. 2 by Supplementary Fig. 9, showing a variance partitioning based on soil variables, topographic variables and radiation for all functional diversity indices and functional traits. This new figure shows the relationship of spatial patterns in functional traits, and especially functional richness, and the environmental gradients of soil and topography more convincingly.

I think this manuscript is moving in the right direction, but much of the continuity has been lost as more analyses have been done but the text has not been appropriately or comprehensively updated.

We revised the text to ensure consistency and continuity throughout the manuscript. By defining three main goals of the study, we better outline the main aspects, relevance and novelty of the study.

Reviewer #2 (Remarks to the Author):

I am feeling that the authors did a great job to reply to my comments together with those of the other reviewers. I am also feeling that some of the comments from Reviewer #1 should still be considered in full detail. E.g. the need of field data to calibrate hyperspectral traits and indices. A more convincing statement should be put in the main text.

We appreciate this suggestion and added a separate subsection 'Field validation of physiological traits' and a specific statement on lines 101-106 and 441-446.

Another minor point concerns the Supplementary material. I am feeling that the frequency distribution of FHD is a key concept in this manuscript which might deserve space in the main ms instead of being basically hidden in the Supplementary material. This is also true for trait correlations. However, this is not only an authors' decision but overall an editorial one.

This said, the paper is robust enough to deserve publication.

Duccio Rocchini

We appreciate the valuation of our work and we are happy that our article is recommended for publication. Since trait correlations are presented in the results section and we would like to emphasize the functional diversity approach in a spatial context, we decided to keep the figures in the supplementary material.

Among Reviewer 3's comments, please focus on the following points:

- **We request that you provide better context in the Introduction and Discussion regarding the importance of understanding functional trait diversity patterns (e.g. the implications for ecosystem function).**

Thank you for this suggestion. We agree that the relevance of studying functional diversity patterns has to be pointed out specifically in the manuscript. We do so now by including an additional paragraph in the introduction citing recent literature on diversity-productivity relationships and the impact of diversity on ecosystem stability. We also discuss potential implications for ecosystem functioning based on the observed diversity patterns. However, to establish a link between functional diversity and ecosystem functioning at our site would exceed the scope of this study.

- **Please consider addressing Reviewer 3's point about correlating forest traits with a detailed soil map, perhaps by making use of the data in Supplementary Figure 10.**

We agree that soil variables are important to explain and potentially predict functional diversity patterns. Therefore, we perform a variance partitioning based on soil variables (soil type, soil depth, amount of coarse grains), topographic variables (altitude, slope, aspect, curvature) and radiation (mean daily photosynthetically active radiation). We add Supplementary Fig. 9, the results on lines 148, 152, 175, 187 and discuss it on lines 228-239, 264-270. We can show that the environmental gradient of changing soil and topography towards the top of the mountain consistently links to the functional richness patterns of morphological and physiological traits.

- **Please establish predictions regarding how functional trait diversity should change with scale (in the context of existing literature on diversity-area relationships).**

We introduce our hypothesis on changing functional trait diversity with increasing area in the introduction. We expect functional richness to increase with scale similarly to species-area relationships. However, the exact shape of the curve cannot be predicted due to intra-specific trait variability, trait plasticity and possible trait correlations. Nevertheless, we found a similar slope of a power law fit in log-log scale than predicted by a large-scale species richness-area model of Gerstner, et al. 2014¹. Furthermore, we found a deviation from the power law at smaller scales, as was discussed in Nature by Pereira, et al. 2011². We added Suppl. Fig. 11 to illustrate this. Divergence and evenness were scale-invariant in our analysis, which is in agreement with Karadimou, et al. 2016³.

Reviewer #3 (Remarks to the Author):

There remain a few conceptual issues, which require further development.

¹ Gerstner, K., et al. Accounting for geographical variation in species-area relationships improves the prediction of plant species richness at the global scale. *Journal of Biogeography* 41, 261–273 (2014).

² Pereira, et al. Geometry and scale in species-area relationships. *Nature* 482, E3–E4 (2012).

³ Karadimou, E. K., et al. Functional diversity exhibits a diverse relationship with area, even a decreasing one. *Scientific Reports* 6, 35420 (2016).

What is meant by functional diversity and what do we learn from it? Here it seems to be the distribution of several traits that are retrievable from RS indices and can then be mapped. But different traits will give different diversity values, and there should be some context and foundation for the choice – linked to larger conceptual questions, presumably about ecosystem function. The functional diversity angle is weak without a clear conceptual framework or link to ecosystem function.

We have rephrased the introduction to better address these issues. We describe the relevance of mapping functional diversity in the context of ecosystem functioning. We also point out the differences and expected similarities between morphological and physiological traits and trait diversity respectively. Although the choice of traits does influence the diversity values, we expect the spatial patterns to converge following broad environmental gradients.

It seems the integration of diversity indices with lidar is the most novel angle rather than “advanced trait retrieval” and mapping traits from airborne data. The relationship between trait maps and forest types is described, but many forest types can be mapped from imagery without trait mapping. Clarification of how the insights in the ms are novel is needed. How is it novel to apply indices of functional diversity to airborne imagery, and what is learned by doing so?

By redefining the main goals of the study in the introduction we clarify the main aspects and the novelty of the study. We agree that the comparison of morphological and physiological diversity derived independently from airborne laser scanning and imaging spectroscopy is novel, especially at this scale and resolution. Therefore we specifically lay out why it is important to compare them. Indeed, the functional trait maps reflect differences between forest types, and by comparing them with community data we can show some consistency (as requested by reviewer 1), but they are not limited to it. By mapping functional traits continuously and deriving functional diversity from it, we can provide a much more direct measure of biodiversity, not limited by given vegetation types or units. We strengthen this point in the manuscript on lines 37-40, 90-95. The novelty is in the continuous large-scale diversity mapping, which does include intra-specific trait variability. This is especially important when studying temperate mixed forests, where intra-specific trait diversity can be as large as inter-specific diversity.

The ms reveals low functional diversity of mountain ridges and high functional diversity of disturbed areas. What does this tell us about ecosystem function? That disturbed and early successional areas are more productive and high altitudes are less productive?

The lower functional diversity on the mountain ridge could indeed indicate that ecosystem functioning is reduced (see lines 233-239 of the Discussion). Disturbed areas do add to the diversity at larger scales, but also have low within-community diversity. Therefore it is more difficult to say if more disturbed areas would lead to higher productivity of the whole forest. It is not the scope of this study though to establish a direct link between functional diversity and ecosystem functioning. There are various recent studies showing a positive relationship between taxonomic / functional diversity and productivity, which we now cite in the introduction.

The relationship between lower functional diversity and higher altitude (ridges) with shallower soils that have fewer resources (nutrients, water) is interesting and could be the dominant focus of the ms. The forest traits may correlate with a detailed soil map, if it exists for the area.

In terms of the way functional diversity is calculated, functional richness, evenness and dispersion are all important components of functional diversity. In addition to the number of functionally distinct units (richness), how different they are (dispersion) and in how much they fill trait space (evenness) matters.

Some concepts in the discussion, including “competitive exclusion”, “underdispersion”, “divergence/convergence” are mentioned but not well explained. The interpretation remains speculative.

As requested by the editor and pointed out in the corresponding answer above, we included soil variables in the analysis and we strengthen the discussion on the relationship between environmental variables, following a gradient with altitude, and trait convergence resulting in lower functional diversity at higher altitudes.

The questions of the manuscript need to have context and to be grounded in a foundation for interpretation. What are the expectation for why trait diversity or dispersion patterns should change with scale and what do we learn when the do or do not shift according to expectations? As posed, three questions set the study up to be fairly descriptive.

1) What are the spatial patterns of morphological and physiological trait diversity derived from remotely sensed laser scanning and spectrometer data?

2) How does functional diversity change with scale?

3) Are most of the forest communities structured by trait convergence, trait divergence or randomly distributed traits?

We need to understand why shifts will scale are important and what the expectations are for the functional diversity to vary with scale and how this relates to the ability of remote sensing to detect functional diversity shifts with scale.

Thank you for raising these important questions. First of all, we agree that the initial questions were descriptive and did not reveal all the relevance and novelty of our study. Therefore, we decided to slightly change the format and clearly define three main goals of the study in the introduction. By doing so, we also provide the relevant context and formulate our expectations e.g. for the change of diversity with scale.

Line 45- 46: Some imprecision in language: “different species can be redundant with regard to their functional diversity” - does this really mean species can be redundant in terms of their functional traits?

Exactly, we rephrased the sentence on line 44.

Lines 50-55: “Canopy height, density, layering ... influence light availability, resource consumption and species diversity species diversity”: Some clarification required here - in the understory? of forest birds? small mammals? what is being referred to?

We rephrased this paragraph, including the referred sentence on lines 63-67.

Line 64: “Quantifying functional diversity from morphological and physiological traits reveals the distribution of species or individuals in the functional trait space”. This is not actually true. The authors intend meaning here that is not being conveyed.

We rephrased the beginning of the paragraph to correct this issue starting on line 77.

Line 66: “Functional richness is calculated as the convex hull volume of the community niche”: This does not account for the filling of the convex hull. Richness would be the number of functional “units”, and the number of units is independent of the niche volume. Many different units can occupy a small space, but three extreme units can delineate a large space. This is why dispersion and evenness are important in FD. Then one could ask whether all three factors (richness, evenness, dispersion) are needed to explain a certain ecosystem function. This would be interesting avenue for the ms.

It is important to clarify that in our approach the “functional units” are pixels, and not individuals, species, or functional types of trees or vegetation. Therefore the number of units used to calculate the functional diversity measures does not vary for a given scale. We use the definition of functional richness of Mouillot, et al. 2013⁴ and Villéger, et al. 2008⁵. Nevertheless, it is true that this functional richness measure does not account for the filling of the convex hull, this is why we also use functional divergence and evenness in addition. This is explained in detail in the paragraph on lines 77-95.

⁴ Mouillot, D., et al. A functional approach reveals community responses to disturbances. *Trends in Ecology & Evolution* 28, 167–177 (2013).

⁵ Villéger, S., et al. New multidimensional functional diversity indices for a multifaceted frame work in functional ecology. *Ecology* 89, 2290–2301 (2008).

Lines 84-86: Why do similar pattern in traits indicate that they are mapped correctly? Is this because all of the traits are correlated with each other? "and are representative for functional traits in general" I am not sure what this is supposed to mean. Certainly not all functional traits are correlated with each other.

We rephrase the importance of comparing the morphological and physiological diversity maps and testing the consistency of our method on lines 96-106. The agreement between our completely independently acquired morphological and physiological metrics is however a strong indicator for the robustness of the derived traits.

Lines 86-86 "Nevertheless, we also expect to see differences due to different responses to abiotic factors and expressions of plant health and development" Here is where a conceptual framework laying out expectations is important. What kinds of patterns and deviations are expected and how should they change with scale associated with the processes and functions that predominate at different scales? This would provide a means to interpret otherwise descriptive patterns.

We reformulate this statement in the introduction to be more precise and present a context of what we expect based on literature (lines 97-100). Then we specifically discuss the differences on lines 240-257 of the discussion.

Lines 126-128: "Patterns of morphological and physiological richness exhibit strongest correlation at medium scale between 60 and 240 m radius." Why is the correlation important to test and what would the expectation be?

We build a stronger argumentation in the introduction, why the comparison and related correlations are important. See lines 95-101.

Lines 280-283: "Functional divergence and evenness are generally high, mainly scale invariant and vary only in a small range, scale-dependency of functional diversity in this forest ecosystem is best represented by functional richness." Without using a functional diversity metric that incorporates all three components and allows examination of the influence of each component separately, this conclusion may not be supported. Or one could investigate the influence of each component of functional diversity and its association with a specific abiotic factor or ecosystem function.

We agree that this conclusion is weak. We decided that it should not be drawn based on the proposed analysis. Instead, we now demonstrate the association of each component with abiotic factors such as soil, topography and radiation.

REVIEWERS' COMMENTS:

Reviewer #1 (Remarks to the Author):

Overall I am happy with this manuscript and how it has improved with revisions. My one remaining question is with the new statistical analyses and Supplementary Figure 9 - the rebuttal explains this figure as variance partitioning but the description in the methods (lines 541-550) is very cursory. Hopefully a few sentences or references to this type of analysis will clear up confusion, but my impression is that no model selection was done to rule out insignificant predictors or to address multicollinearity, but depending on how the models were constructed (top 2 panels vs bottom 2) the strength of the predictors varied? That seems problematic, along with the lack of testing for spatial autocorrelation... given that this is a relatively small component of the overall paper, and I don't think it would change the points made in the discussion, I would just scale this back to look at individual correlations between the predictors and the traits (using a modified t-test approach to test for significance while considering spatial autocorrelation like that described by Dutilleul et al. (1993)).

Dutilleul, P. et al. 1993. Modifying the t test for assessing the correlation between two spatial processes. – *Biometrics* 49:305–314.

Reviewer #3 (Remarks to the Author):

I find the manuscript quite strong. The authors are to be lauded for a tremendous amount of work, a novel, integrative approach, and much additional effort in the course of this lengthy review process.

Here are a series of small wording edits for clarity/readability:

line 10: change "could help predicting" to "can help predict"

line 18: change "mixtures of tree functional groups" to "composition of tree functional groups"

line 37: You might want to add Williams et al 2017 *Nature EE* in reference to complementarity effects

line 46-48: suggest changing: "Incorporating individual-level functional traits, functional diversity may better predict ecosystem functioning than only using species level means can do"

to

"By incorporating individual-level functional traits, functional diversity may better predict ecosystem functioning than species level means."

line 230: I suggest replacing the word "convergence". The meaning is not clear.

line 246: "ecosystem functioning might be increased" is too vague. You would have to identify the function in order to explain that it increased.

lines 277 & 278; the sentence need some adjustment in construction: Given the continuous nature of the remotely sensed functional trait maps, we were able to study functional diversity at multiple scales and to develop a highly resolved scaling relationship.

Conclusions:

line 320. Delete "uniquely". Meaning is unclear and term is not necessary.

line 330. Using the term "convergence" is not appropriate because that is an evolutionary term meaning that traits evolved to be similar due to similar environmental selection pressures. "low variance", "homogeneity" or "similarity" would be ecological terms.

line 335. Can you avoid using "should" in the sentence? Also, ecosystem functioning appears twice. Simplify.

How about:

Future studies can advance the integration of remotely sensed functional data with databases of plant functional traits, environmental and ecosystem data, and dynamic vegetation models to increase our understanding of the mechanistic linkages between functional diversity and ecosystem function.

REVIEWERS' COMMENTS:

Reviewer #1 (Remarks to the Author):

Overall I am happy with this manuscript and how it has improved with revisions. My one remaining question is with the new statistical analyses and Supplementary Figure 9 - the rebuttal explains this figure as variance partitioning but the description in the methods (lines 541-550) is very cursory. Hopefully a few sentences or references to this type of analysis will clear up confusion, but my impression is that no model selection was done to rule out insignificant predictors or to address multicollinearity, but depending on how the models were constructed (top 2 panels vs bottom 2) the strength of the predictors varied? That seems problematic, along with the lack of testing for spatial autocorrelation... given that this is a relatively small component of the overall paper, and I don't think it would change the points made in the discussion, I would just scale this back to look at individual correlations between the predictors and the traits (using a modified t-test approach to test for significance while considering spatial autocorrelation like that described by Dutilleul et al. (1993)).

Dutilleul, P. et al. 1993. Modifying the t test for assessing the correlation between two spatial processes. – Biometrics 49:305–314.

We would like to thank the reviewer for detailed comments. We agree that spatial autocorrelation needs to be considered in the statistical analysis. Therefore, we now fit a spatial model and use the estimated covariance matrix to fit a generalized linear model in order to account for spatial dependencies based on first order neighbors. We use the R package *spdep* and the function *errorsarm* to compute relevant statistical figures (cf Bivand and Piras, 2015¹). Then we perform an ANOVA type I. Supplementary Figure 4 shows now the variance explained based on type-I sum of squares by soil (top panels) and topography (bottom panels), as well as what is additionally explained by adding topography or soil, respectively, and radiation to the model. Within the groups, the order of the explanatory variables was kept constant. For Supplementary Tab. 1, the order of the explanatory topographic variables was determined by the significance when tested individually, with the most significant used first in the combined model. We added this description and the details about the spatial model to the Methods (Statistical analysis, lines 517-534). We performed model selection and found a linear model to be best suited for the analysis. However, we did not aim to exclude any explanatory variable, since we only used variables of interest in the model without including any nuisance variables.

¹ Bivand, R. & Piras, G. Comparing Implementations of Estimation Methods for Spatial Econometrics. *Journal of Statistical Software* **63** (2015).

Reviewer #3 (Remarks to the Author):

I find the manuscript quite strong. The authors are to be lauded for a tremendous amount of work, a novel, integrative approach, and much additional effort in the course of this lengthy review process.

We appreciate the valuation of our work and value much those comprehensive comments.

Here are a series of small wording edits for clarity/readability:

line 10: change "could help predicting" to "can help predict"

Change applied on line 10.

line 18: change "mixtures of tree functional groups" to "composition of tree functional groups"

We removed the whole sentence based on an editorial comment.

line 37: You might want to add Williams et al 2017 Nature EE in reference to complementarity effects

Thank you for this suggestion, we added the reference to the manuscript (line 37).

line 46-48: suggest changing: "Incorporating individual-level functional traits, functional diversity may better predict ecosystem functioning than only using species level means can do" to

"By incorporating individual-level functional traits, functional diversity may better predict ecosystem functioning than species level means."

Change applied on lines 46-48.

line 230: I suggest replacing the word "convergence". The meaning is not clear.

We replaced the word by "reduced trait variability" (line 225).

line 246: "ecosystem functioning might be increased" is too vague. You would have to identify the function in order to explain that it increased.

We rephrased the sentence on lines 240-243.

lines 277 & 278; the sentence need some adjustment in construction: Given the continuous nature of the remotely sensed functional trait maps, we were able to study functional diversity at multiple scales and to develop a highly resolved scaling relationship.

Thank you for this comment. We applied these changes on lines 272-273.

Conclusions:

line 320. Delete "uniquely". Meaning is unclear and term is not necessary.

Change applied on line 307.

line 330. Using the term "convergence" is not appropriate because that is an evolutionary term meaning that traits evolved to be similar due to similar environmental selection pressures. "low variance", "homogeneity" or "similarity" would be ecological terms.

We removed this sentence when merging Conclusions with Discussion, based on an editorial comment.

line 335. Can you avoid using "should" in the sentence? Also, ecosystem functioning appears twice. Simplify.

How about:

Future studies can advance the integration of remotely sensed functional data with databases of plant functional traits, environmental and ecosystem data, and dynamic vegetation models to increase our understanding of the mechanistic linkages between functional diversity and ecosystem function.

Thank you for the suggestion. We applied the changes on lines 310-313.